# Dissociable roles of the inferior longitudinal fasciculus and fornix in face and place perception

**Carl J Hodgetts**[1,2,3]*†, **Mark Postans**[1,2,3]†, **Jonathan P Shine**[1,2,3], **Derek K Jones**[1,2,3], **Andrew D Lawrence**[1,2,3]*, **Kim S Graham**[1,2,3]*

[1]School of Psychology, Cardiff University, Cardiff, Wales; [2]Wales Institute of Cognitive Neuroscience, School of Psychology, Cardiff University, Cardiff, Wales; [3]Neuroscience and Mental Health Research Institute, Cardiff University, Cardiff, Wales

**Abstract** We tested a novel hypothesis, generated from representational accounts of medial temporal lobe (MTL) function, that the major white matter tracts converging on perirhinal cortex (PrC) and hippocampus (HC) would be differentially involved in face and scene perception, respectively. Diffusion tensor imaging was applied in healthy participants alongside an odd-one-out paradigm sensitive to PrC and HC lesions in animals and humans. Microstructure of inferior longitudinal fasciculus (ILF, connecting occipital and ventro-anterior temporal lobe, including PrC) and fornix (the main HC input/output pathway) correlated with accuracy on odd-one-out judgements involving faces and scenes, respectively. Similarly, blood oxygen level-dependent (BOLD) response in PrC and HC, elicited during oddity judgements, was correlated with face and scene oddity performance, respectively. We also observed associations between ILF and fornix microstructure and category-selective BOLD response in PrC and HC, respectively. These striking three-way associations highlight functionally dissociable, structurally instantiated MTL neurocognitive networks for complex face and scene perception.

*For correspondence: hodgettscj@cardiff.ac.uk (CJH); lawrencead@cardiff.ac.uk (ADL); grahamks@cardiff.ac.uk (KSG)

†These authors contributed equally to this work

Competing interests: The authors declare that no competing interests exist.

## Introduction

Recent studies suggest that substructures in the medial temporal lobe (MTL) contribute to processes beyond memory, with the hippocampus (HC) and perirhinal cortex (PrC) necessary for accurate perceptual discrimination of conjunctive scene and face stimuli, respectively (*Graham et al., 2010*). It has been shown, for example, that HC damage leads to impairments in perceiving, learning, and remembering complex scenes, whereas broader MTL damage, affecting both the HC and PrC, results in perceptual and mnemonic deficits for scene, but also face, stimuli (*Barense et al., 2005*; *Lee et al., 2005b*; *Mundy et al., 2013*). Critically, however, these same patients are able to perform perceptual discriminations on the basis of simple visual features (e.g., size and colour) and can learn to discriminate between complex dot patterns (*Mundy et al., 2013*). These findings have been complemented by functional MRI (fMRI) studies showing (a) differential recruitment of the HC and PrC for scene and face discriminations, respectively, including on tasks analogous to those affected in lesion patients (*Barense et al., 2010*), and (b) modulations of blood oxygen level-dependent (BOLD) response by visual feature overlap in both MTL (high > low) and occipito-temporal (low > high) regions (*Mundy et al., 2012*).

This body of work challenges the idea that the MTL is involved exclusively in mnemonic processes (*Squire et al., 2007*) and highlights the need to understand how different types of representational content may drive recruitment of the HC and PrC in perceptual, but also memory, tasks (*Graham et al., 2010*). Emerging hierarchical, representational theories provide a framework for such investigations

**eLife digest** Perceiving an object or picture stimulates activity in the regions of the brain that make up the visual system. Some of these regions respond differently depending on what is being viewed: for example, some areas are more active when looking at faces, and others respond more when viewing places. One theory is that, rather than working in a self-contained fashion, category-sensitive brain regions are elements or 'nodes' within more complex brain networks that are specialised for processing different types of visual stimuli.

The inside of the brain contains regions of dark and light tissue. The lighter regions are known as 'white matter' and contain fibres that allow information to travel between different parts of the brain. These fibers may play an important role in how widely distributed brain regions communicate. To investigate this, Hodgetts, Postans et al. used a technique called diffusion MRI to measure the structure, or coherence, of white matter fibers in healthy volunteers. Brain activity was also measured while volunteers completed a task in which they needed to spot the odd-one-out from images of either faces or places.

Hodgetts, Postans et al. investigated the fine structure of a white matter fibre bundle known as the inferior longitudinal fasciculus (ILF). This fibre links two parts of the brain involved in face perception, called the occipital and anterior temporal lobes. Strikingly, ILF structure predicted both face-related brain activity in these regions and how well an individual could discriminate between faces, but not place stimuli.

By contrast, the ability of volunteers to tell apart different places (but not faces) was related to the structure of the fornix. The fornix is a bundle of white matter fibres that carries information to and from the hippocampus, a region that is important for finding one's way around an environment and remembering such journeys afterwards.

Hodgetts, Postans et al.'s findings suggest that the systems that process different visual categories are best thought of as large-scale distributed networks rather than a set of individual, specialised regions in the brain. In the future, research will be needed to further understand how white matter contributes to the perception of different visual categories, and to investigate in finer detail how visual experience influences the structure of white matter pathways.

(*Graham et al., 2010*; *Saksida and Bussey, 2010*); they propose that perceptual difficulties in patients with MTL involvement reflect damage to conjunctive scene and face/object representations stored within the HC and PrC, respectively (*Murray et al., 2007*; *Graham et al., 2010*; *Saksida and Bussey, 2010*). Any task, whether perceptual or mnemonic, that places demand on these conjunctive representations, such as discriminating between exemplars with many overlapping features (*Saksida and Bussey, 2010*), is predicted to lead to impairment. By contrast, processing of visual exemplars with minimal overlap can be supported by posterior visual cortical regions (*Barense et al., 2010*; *Mundy et al., 2012*).

The strongest evidence for these accounts has come from studies using oddity judgement tasks in which participants detect an 'odd-one-out' stimulus from an array of same-category items presented from different viewpoints (*Lee et al., 2005a*, *2006*). For example, macaque monkeys with lesions to the PrC are impaired on oddity tasks when presented with face and object arrays (*Buckley et al., 2001*); this pattern is also seen in amnesic patients when damage to the MTL includes PrC (*Lee et al., 2005a*). Conversely, patients with selective damage to the HC perform as well as controls on face and object oddity judgements, but show impairments for scene oddity decisions (*Lee et al., 2005a*).

While studying the functional dissociation between MTL regions has been revealing, a fundamental question is how these distinct representations might emerge via wider, distributed, and interactive brain networks (*Mesulam, 1990*, *1994*). More specifically, it has been proposed that, 'functional specialization is not simply an intrinsic property of individual regions that compute specific representations in isolation, but rather, is an emergent property of the interactions between a set of spatially distributed nodes and their functional and structural connections' (*Behrmann and Plaut, 2013*, p. 211).

The fornix—a major input and output pathway of the HC—has long been considered to form part of an extended hippocampal system, both in terms of its role in episodic memory (*Aggleton and Brown, 1999*;

*Metzler-Baddeley et al., 2011*), but also in spatial processing (*Bird and Burgess, 2008*). For example, the impact of fornix transection in non-human primates suggests that exchange of information between HC and diencephalic regions is necessary for learning both object-in-place associations and conjunctions of spatial features (*Gaffan, 1994*; *Buckley et al., 2004*). Fornix damage in humans also impairs episodic memory, as assessed by standardised neuropsychological tasks (*Tsivilis et al., 2008*), and fornix microstructure has been shown to correlate with scene recollection (*Rudebeck et al., 2009*), suggesting a potential convergence between findings in human and non-human primates.

While these studies support the notion that the HC-diencephalic connection established by the fornix is important in memory, there is no evidence that inter-individual variability in this pathway in humans is associated with spatial *perception*, specifically where tasks modified from the animal literature are applied (*Buckley et al., 2001*), and where mnemonic demands are minimised through use of concurrent stimulus presentation and trial-unique stimuli. If the HC contains conjunctive scene representations that can sub-serve both scene memory and perception, as predicted by representational accounts, then inter-individual variability in fornix tissue microstructure may partly support interactions between the HC and its interlinked brain regions within the limbic system that, in concert, underlie successful scene discrimination.

A second relevant white matter (WM) pathway is the inferior longitudinal fasciculus (ILF). This tract serves as the primary input pathway to the antero-medial temporal lobe, including PrC (*Catani et al., 2003*), and is comprised mainly of long association fibres connecting extrastriate visual areas with regions in ventral anterior temporal (vATL) cortex (*Latini, 2015*). Representational accounts propose that brain regions within the antero-medial temporal lobe are part of an extended representational system in the visual ventral stream (*Murray et al., 2007*; *Saksida and Bussey, 2010*); in these hierarchical views, object discriminations requiring complex and conjunctive visual representations are supported by the PrC and those based on lower-level perceptual features are dependent on early visual areas (*Mundy et al., 2012, 2013*). Given this, microstructural properties of the ILF—a major WM tract of this extended visual stream (*Mishkin et al., 1983*; *Yeterian and Pandya, 2010*)—could influence representations in PrC that are required to differentiate between complex object and/or face stimuli. Indeed, there is evidence suggesting that ILF microstructure is related to performance in tasks involving face stimuli; for example, individuals with congenital prosopagnosia (CP, a syndrome characterised by impairments in identifying faces) have altered WM microstructure and macrostructure (i.e., volume) relative to matched controls in ILF (*Thomas et al., 2009*; *Gomez et al., 2015*). These studies suggest that the ILF may be a key tract in a network supporting the perceptual processing of face stimuli and may, through its antero-medial temporal connections, mediate performance in conjunctive face discriminations impacted directly by damage to vATL regions, such as PrC (*Lee et al., 2005a, 2006*).

These studies imply, therefore, that the ILF and fornix may underpin structurally distinct distributed neural circuits that are specialised for information processing of different types of visual representations. Unfortunately, however, such a conclusion is limited by two key weaknesses in the literature. First, there has been no *double dissociation* of the functional contributions of the ILF and the fornix within the same participants. Consequently, it is possible that any purported cognitive dissociations reflect the different tasks and/or methodological approaches used to investigate the fornix/ILF independently. Second, no study has yet manipulated representational content within the same experimental paradigm where cognitive demand and difficulty were also matched, as well as ensuring provision of a stringent control condition in which no statistical association is predicted with either WM tract.

Here, we addressed these issues by using diffusion tensor imaging (DTI) in healthy individuals to test our primary hypothesis that the fornix and ILF are differentially associated with complex scene and face perception, respectively. This prediction was based on the distinct structural connections these tracts establish with the HC and vATL, and generated from published theoretical/computational accounts of representational models (*Cowell et al., 2010*; *Elfman et al., 2014*). To draw correspondence with lesion studies in animals and humans, and to provide the strongest test of representational accounts, we used a modified odd-one-out task in which mnemonic demand was minimised (see 'Materials and methods'). By applying diffusion tractography methods, we extracted two microstructural measures: mean diffusivity (MD) and fractional anisotropy (FA). MD ($10^{-3}$ mm$^2$ s$^{-1}$) reflects a combined average of axial diffusion (diffusion along the principal axis) and radial diffusion (diffusion along the orthogonal direction), and FA reflects the extent to which diffusion within biological tissue is anisotropic, or constrained along a single axis, and can range from 0 (fully isotropic) to 1 (fully anisotropic). Decreases in MD (and also increases in FA) are associated, typically, with microstructural properties that are

considered to support the efficient transfer of information along WM, such as increased myelination and axon density (*Beaulieu, 2002*; but see; *Jones et al., 2013*). Based on this, we predicted that individual success on scene oddity would correlate negatively with fornix MD and positively with fornix FA. Conversely, we predicted that accuracy in the face oddity condition would correlate negatively with ILF MD and positively with ILF FA. Furthermore, we hypothesised that measures obtained from these pathways would not be associated with performance on an equally difficult size oddity control condition, which is unaffected following damage to MTL (*Buckley et al., 2001*).

A second analysis investigated the relationship between HC and PrC BOLD response during oddity judgement and task accuracy. Based on a potential representational hierarchy along the ventral visual stream (*Murray et al., 2007*; *Saksida and Bussey, 2010*), we also studied the face-selective ventral occipitotemporal fusiform cortex (fusiform face area, FFA). This region is known to receive ILF inputs (*Gschwind et al., 2012*; *Pyles et al., 2013*), and in combination with the PrC, supports complex face perception (*O'Neil et al., 2012, 2014*). In this analysis, we predicted that inter-individual variation in BOLD response (in a category-specific manner, within the key regions-of-interest, ROIs) would be associated with oddity performance.

Finally, we used a mediation analysis (*Hayes, 2013*) to examine the three-way relationship between regional functional activity, WM microstructure, and oddity performance. If broader MTL neural circuits support complex visual discrimination, then the relationship between regional activity in HC and PrC/FFA and scene and face oddity accuracy, respectively, may be partially mediated by inter-individual variation in fornix and ILF WM microstructure, respectively.

## Results

Both diffusion-weighted and fMRI data were acquired in 30 healthy adult participants who completed a perceptual oddity task for scenes and faces (and a size oddity control condition). To ensure that mnemonic demand was minimised in the oddity task, stimuli were presented concurrently on each trial and never repeated, once shown, in the task (*Lee et al., 2005a*). Performance in this paradigm was defined as the proportion of trials for which participants selected the correct odd-one-out stimulus from the array of three stimuli (see 'Materials and methods'). Two of these stimuli were the same item from different viewpoints, and the third stimulus was a different, but similar, item (*Figure 1*).

### Behaviour

To ensure that behavioural performance (proportion correct) was matched across stimulus types, and that no learning was shown across task runs, we conducted a 3 (task: scenes, faces, size) × 2 (run number: run 1, run 2, run 3) analysis of variance (ANOVA). This analysis revealed no effect of task (F (2, 28) = 1.47, p = 0.24) or run number (F (2, 28) = 1.37, p = 0.26) on behavioural performance and no significant interaction between these two factors (p = 0.27). These results indicate that performance was matched across the three conditions and that there was no improvement in accuracy across task runs (i.e., learning). Descriptive statistics for the behavioural task are shown in *Table 1*.

### DTI and behaviour

#### Tractography

Based on the directional hypotheses outlined above, we conducted one-tailed correlations between free water corrected MD and FA values of the ILF and fornix (obtained separately for each participant) and individual accuracy on oddity judgements for faces, scenes, and size (see 'Materials and methods' for further information). As one participant's mean FA value for the fornix was lower than three standard deviations below the group mean (SD < 0.25), they were removed in order to reduce the effect of outliers on the subsequent correlational analyses. All analyses reported below, therefore, are based on the remaining 29 participants. The mean and standard deviation for each tract (ILF, fornix) and DTI metric (MD, FA) are shown in *Table 2*. Correlations were Bonferroni corrected by dividing the 0.05 alpha level by the number of statistical comparisons for each individual DTI measure (i.e., 0.05/3 = 0.017). The 95% confidence interval (CI) for each correlation was derived using a bootstrapping procedure based on 1000 iterations. Individual FA and MD values for each participant (and tract) are available in *Table 2—source data 1*. A summary table of the DTI-behaviour correlations can be found in *Table 3*.

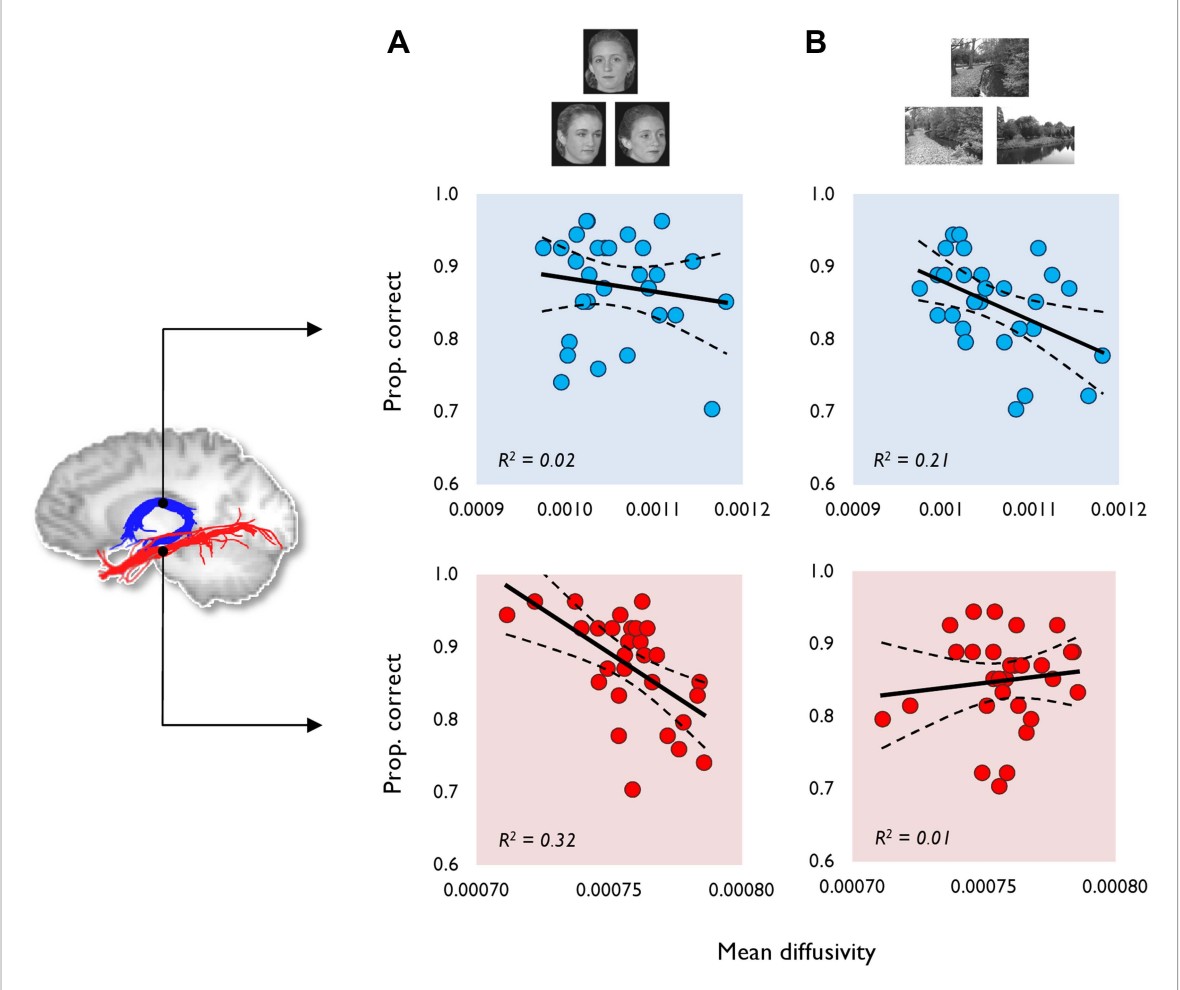

**Figure 1**. The relationship between tract MD and oddity performance (proportion correct). This is shown for (**A**) face and (**B**) scene oddity. The best fitting linear regression line and 95% confidence interval (CI) are displayed on each scatter plot. Reconstructions of the fornix (blue) and inferior longitudinal fasciculus (ILF) (red) are depicted on the sagittal midline slice of a participant's T1-weighted brain image. Data obtained from the fornix and ILF are indicated with blue and red data markers, respectively, with 29 data points appearing on all graphs. Scatter plots for both the fractional anisotropy (FA) data and the size oddity condition are depicted in *Figure 1—figure supplement 1*.

The following figure supplement is available for figure 1:

**Figure supplement 1**. Additional results for the diffusion tensor imaging (DTI)-behaviour correlational analyses.

Face oddity performance was, as predicted, negatively associated with ILF MD (*Figure 1A*). This relationship was found to be significant at the experiment-wise significance level ($r = -0.57$, $p = 0.00$, 95% CI [$-0.76$, $-0.36$]). ILF MD did not correlate significantly with scene ($r = 0.12$, $p = 0.27$; *Figure 1A*) or size oddity performance ($r = -0.26$, $p = 0.09$; *Figure 1—figure supplement 1*). While a positive correlation was observed between ILF FA and face odd-one-out accuracy, this was not statistically significant ($r = 0.30$, $p = 0.06$, 95% CI [0.62, $-0.1$]; *Figure 1—figure supplement 1*). ILF FA did not correlate significantly with individual scores in either the scene ($r = 0.04$, $p = 0.43$) or size oddity ($r = 0.09$, $p = 0.32$) conditions.

To determine whether the ILF correlations obtained for the face condition were significantly greater than for the scene condition, these were compared using a directional Steiger Z-test for dependent correlations (*Lee and Preacher, 2010*), which takes into account the within-subject correlation between the compared task conditions (face vs scene oddity, $r = 0.19$). The difference between the face-ILF MD and the scene-ILF MD correlations was found to be statistically significant

**Table 1**. Descriptive statistics for the three behavioural conditions

|  | Accuracy | | RT | |
|---|---|---|---|---|
|  | *Mean* | *SD* | *Mean* | *SD* |
| Face | 0.87 | 0.07 | 2932.40 | 479.68 |
| Scene | 0.85 | 0.06 | 3147.86 | 557.49 |
| Size | 0.84 | 0.12 | 2464.10 | 668.31 |

Mean and standard deviation (SD) are reported for both accuracy (proportion correct) and reaction time (RT). Raw behavioural data for the three oddity categories are available in *Table 1—source data 1*.

**Source data 1**. Raw behavioural data from the oddity task.

**Table 2**. Descriptive statistics for the fornix (left) and inferior longitudinal fasciculus (ILF, right)

|  | Fornix | | ILF | |
|---|---|---|---|---|
|  | *Mean* | *SD* | *Mean* | *SD* |
| FA | 0.373 | 0.031 | 0.429 | 0.023 |
| MD | 1.058 | 0.054 | 0.758 | 0.017 |

Mean and standard deviations are reported for fractional anisotropy (FA) and mean diffusivity (MD $\times 10^{-3}$ mm$^2$ s$^{-1}$). Individual FA and MD values for each participant (and tract) are available in *Table 2—source data 1*.

**Source data 1**. Raw values for the DTI metrics.

($z$ (26) = 2.07, p = 0.00). For ILF FA, there was no significant difference between the scene and face oddity correlations (p = 0.14).

Based on previous studies that have observed a right hemispheric WM dominance in face processing (*Thomas et al., 2009*; *Gschwind et al., 2012*; *Scherf et al., 2014*), we compared the relationship between face oddity performance and microstructural variation in left and right ILF. While a numerically stronger relationship was observed in the right ILF, there was a significant negative relationship between ILF MD and face oddity performance in both the right ($r = -0.61$, p = 0.00, 95% CI [−0.77, −0.38]) and left hemisphere ($r = -0.43$, p = 0.01, 95% CI [−0.74, −0.06]). Scene oddity success did not correlate with either left (p = 0.20) or right (p = 0.42) ILF. Likewise, there were no significant correlations between size oddity performance and ILF MD in either hemisphere (left ILF: p = 0.17; right ILF: p = 0.06).

In contrast to the ILF MD data, there was a stronger trend between face oddity performance and FA in the left ILF, though this did not reach statistical significance ($r = 0.33$, p = 0.04, 95% CI [0.63, −0.00]). For right ILF FA, there was only a slight positive correlation with face oddity accuracy ($r = 0.18$, p = 0.18, 95% CI [0.48, −0.22]). There were no discernible associations between scene and size oddity and ILF FA in either hemisphere (all *ps* > 0.25). A Steiger Z-test between left and right ILF revealed no significant differences for MD ($z$ (26) = 1.30, p = 0.1) or FA ($z$ (26) = 0.76, p = 0.22). Overall, these hemispheric comparisons confirmed significant correlations between left and right ILF MD and face oddity performance, with a numerically stronger association in the right ILF, consistent with a relative right hemisphere specialisation for face perception (*Behrmann and Plaut, 2013*).

**Table 3**. Summary table for the DTI-behaviour correlations

|  | Fornix | | | | ILF | | | |
|---|---|---|---|---|---|---|---|---|
|  | MD | | FA | | MD | | FA | |
|  | *r* | *p* | *r* | *p* | *r* | *p* | *r* | *p* |
| Face | −0.14 | 0.23 | 0.15 | 0.22 | −0.57 | 0.00 | 0.30 | 0.06 |
| Scene | −0.46 | 0.01 | 0.36 | 0.03 | 0.12 | 0.27 | 0.04 | 0.43 |
| Size | −0.29 | 0.06 | 0.18 | 0.18 | −0.26 | 0.09 | 0.09 | 0.32 |

Correlation coefficients (and one-tailed p values) are reported for each metric (fractional anisotropy [FA], mean diffusivity [MD]) of fornix and inferior longitudinal fasciculus (ILF) microstructure and each task condition (face, scene, size).
DTI, diffusion tensor imaging.

Complementing the ILF data, a significant negative correlation (*Figure 1B*) was observed between fornix MD and scene oddity performance ($r = -0.46$, p = 0.01, 95% CI [−0.69, −0.13]; *Figure 1B*). There was no significant correlation between fornix MD and face oddity ($r = -0.14$, p = 0.23). The correlation between fornix MD and size oddity was likewise not significant ($r = -0.29$, p = 0.06; *Figure 1—figure supplement 1*). The comparison between scene oddity accuracy and FA ($r = 0.36$, p = 0.03, 95% CI [0.67, −0.05]) did not reach the required experiment-wise significance level (*Figure 1—figure supplement 1*). Fornix FA did not correlate with face ($r = 0.15$, p = 0.22) or size ($r = 0.18$, p = 0.18) oddity performance.

Using the directional Steiger Z-test, we found, for fornix MD, that there was a marginally

significant difference between the two correlations obtained for scenes and faces (z (26) = 1.39, p = 0.08). There was no significant difference between the scene and face correlations for fornix FA (p = 0.18).

## Comparisons with size oddity

While none of the microstructural measures obtained, in either pathway, were significantly associated with performance in the difficulty-matched size oddity condition, there were, as reported above, small-to-moderate one-tailed trends between fornix/ILF MD and size oddity (*Figure 1—figure supplement 1*). A Steiger Z-test comparing these coefficients revealed a significant difference between the face and size oddity correlation for ILF MD (z (26) = 2.05, p = 0.02). The difference between the size and scene oddity correlations for fornix MD did not differ significantly (z (26) = 0.94, p = 0.17).

We also conducted partial correlations to see whether the significant relationship between face/scene oddity and ILF/fornix MD remains when size oddity is controlled for, that is, to show that WM microstructure is predictive of face/scene oddity over and above its contribution to lower-level visual discriminations. When size oddity was controlled for, we still observed significant associations (one-tailed) between scene oddity and fornix MD (r = −0.38, p = 0.02, 95% CI [−0.61, −0.08]) and face oddity and ILF MD (r = −0.53, p = 0.00, 95% CI [−0.61, −0.08]).

## Tract-based spatial statistics (TBSS)

To complement the tractography analysis, we also conducted a whole-brain TBSS analysis to investigate any potential voxel-wise correlations outside our main ROIs for scene vs face oddity (S > F) and vice versa (F > S; see 'Materials and methods' for further details). While there were no whole-brain WM voxels showing a significantly greater association with S > F (for either MD or FA), the F > S contrast revealed 23 whole-brain clusters for MD, which were predominantly located in the right hemisphere (see *Supplementary file 1*). The peak of the largest cluster was located in right superior longitudinal fasciculus (SLF; 31, −33, 37; p = 0.02) and incorporated right anterior ILF and the right inferior fronto-occipital fasciculus (IFOF; *Figure 2*). Sub-peak clusters were found in left corpus callosum, right middle ILF, and the left middle portion of the cingulum bundle. There were 53 significant voxels located in the ILF bilaterally with the peak located in the right hemisphere (44, −8, −18; p = 0.03).

For FA, nine clusters were identified that were more strongly correlated with faces than scenes. The peak for the largest cluster was located in the right callosal body (16, 12, 29; p = 0.00) and incorporated several prominent WM structures bilaterally, including ILF, SLF, and forceps major (see *Supplementary file 1*). A second cluster of 118 voxels was identified with a peak in the right anterior ILF (40, −33, −15; p = 0.05). There were 250 significant voxels in ILF with the peak located in the left ILF. A figure displaying the TBSS clusters can be found in *Figure 2*.

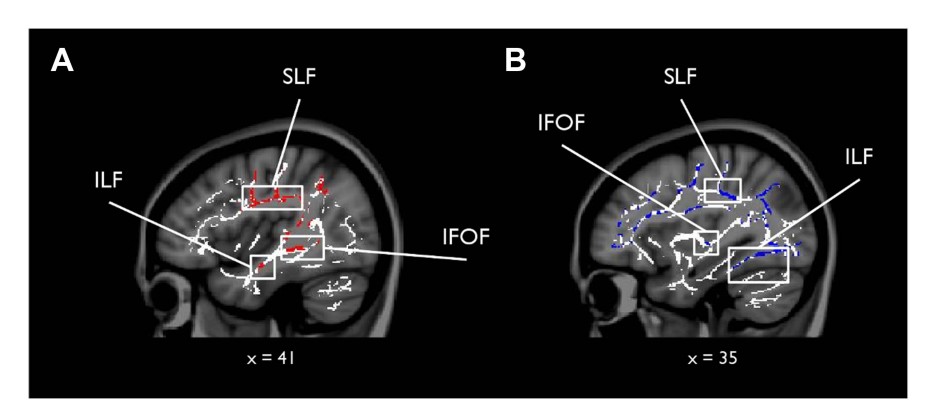

**Figure 2**. Results of the whole-brain tract-based spatial statistics (TBSS) analysis. White matter (WM) voxels identified by TBSS that show a stronger relationship for F > S and (**A**) mean diffusivity (MD) (negative correlation, red clusters), and (**B**) FA (positive correlation, blue clusters). Abbreviated WM structures include SLF (superior longitudinal fasciculus), IFOF (inferior fronto-occipital fasciculus), and ILF (inferior longitudinal fasciculus). There were no whole-brain WM voxels showing a significantly greater association with S > F (for both MD and FA). A table of peak coordinates for the TBSS analysis can be found in *Supplementary file 1*.

## BOLD and DTI

As MD and FA are associated with properties that affect the efficiency of information transfer along axons (Beaulieu, 2002), it is highly likely that inter-individual variation in these measures will impact on, and constrain, BOLD activity in specific ROIs (Behrens and Johansen-Berg, 2005). Here, we used probabilistic atlases to define bilateral ROIs of PrC, FFA, and HC and employed a general linear model (GLM) to test for voxel-wise linear associations between WM microstructure and category-selective BOLD (see 'Materials and methods'). Given that increased MD reflects greater diffusion along both the axial and radial diffusion directions, we predicted a negative association with BOLD activity. For FA, we predicted a positive association with BOLD, as this metric reflects the extent to which diffusion within biological tissue is highly directional, or constrained along a single axis. The data from four participants were excluded from the analysis due to excessive movement during the functional run (>3 mm), and a further participant removed due to scanner error, resulting in a sample of n = 24 for all subsequent analyses.

As shown in *Figure 3A*, the statistical map reflecting a negative association between inter-subject BOLD for F > S and ILF MD revealed significant bilateral clusters in FFA (*left:* −28, −52, −18, Z = 2.99, 25 voxels; *right:* 44, −52, −18, Z = 3.07, 41 voxels). There was also a cluster of 11 voxels in left PrC (−32, −12, −34, Z = 2.81), which did not quite reach our PrC cluster extent threshold (cluster >17 voxels, p = 0.05; *Figure 3B*). No clusters were found for F > S and ILF FA. As these functional associations with MD could arise from between-subject variability in the scene oddity baseline, we also conducted this analysis for F > rest; as above, we identified significant bilateral clusters in FFA associated with ILF MD, with the larger, stronger, cluster located in right hemisphere (*left:* −28, −52, −18, Z = 2.98, 19 voxels; *right:* 42, −54, −18, Z = 4.21, 210 voxels; *Figure 3A*). We also identified bilateral face-sensitive clusters in the PrC associated with ILF MD (*left:* −32, −14, −34, Z = 3.52, 19 voxels; *right:* 28, −16, −32, Z = 3.52, 27 voxels; *Figure 3B*). There were no supra-threshold clusters in FFA or PrC for ILF FA. Across two BOLD contrasts (F > S and F > rest), therefore, inter-individual differences in ILF microstructure (MD) were correlated with BOLD response to faces in FFA and PrC (*Pyles et al., 2013*).

In the HC, there were no significant clusters for our two scene contrasts (S > F or S > rest) and either measure of fornix microstructure. Given evidence for a decoupling between HC BOLD and underlying neuronal activity (*Ekstrom, 2010*), in particular that negative BOLD changes in the HC are often accompanied by increased spike rate or synaptic input (see 'Discussion' for further details), we also investigated whether fornix MD or FA might be associated with inter-individual differences in scenes compared to rest (S < rest, i.e., task-induced hippocampal deactivations). This analysis revealed a significant cluster in left intermediate HC (−26, −24, −16, Z = 3.1, 32 voxels; *Figure 3C*) that was strongly associated with fornix FA. The fornix MD analysis revealed a cluster in right anterior HC (20, −16, −16, Z = 2.24) albeit at a lower voxel-wise threshold (p = 0.05). To test whether this deactivation effect was specific to scene oddity, we conducted the same analysis for face deactivations against rest baseline (F < rest); this revealed no voxels in HC associated with either fornix MD or FA.

To test whether fornix microstructure is associated with scene-selective BOLD in other scene-selective cortical regions (*Epstein, 2014*), we conducted an additional voxel-wise analysis within anatomically defined, independent ROIs sampling the posterior parahippocampal gyrus (PHG), retrosplenial cortex (RSC), and transverse occipital sulcus (TOS; see 'Materials and methods'). No significant clusters were found that showed a significant positive or negative association between scene-selective BOLD (S > F, S > rest) and fornix microstructure (MD or FA) in any of the additional scene-selective ROIs.

## BOLD and behaviour

We next correlated mean individual percentage BOLD signal change values from each probabilistic anatomical ROI with face and scene oddity performance (one-tailed, *Figure 4*). A significant positive relationship was found between face-sensitive BOLD (F > rest) in FFA and face oddity accuracy (r = 0.36, p = 0.04, 95% CI [0.05, 0.65]; *Figure 4A*). A significant positive correlation was also observed between BOLD for faces (F > rest) in PrC and face oddity performance (r = 0.42, p = 0.02, 95% CI [0.06, 0.69]; *Figure 4B*). Neither ROI was correlated with scene oddity performance (all ps > 0.2).

There was also a strong trend evident between scene-related deactivations (S < rest) and scene oddity accuracy in the HC ROI (r = 0.30, p = 0.08, 95% CI [−0.039, 0.606]; *Figure 4C*); an association that was not evident for the face oddity condition (p = 0.20).

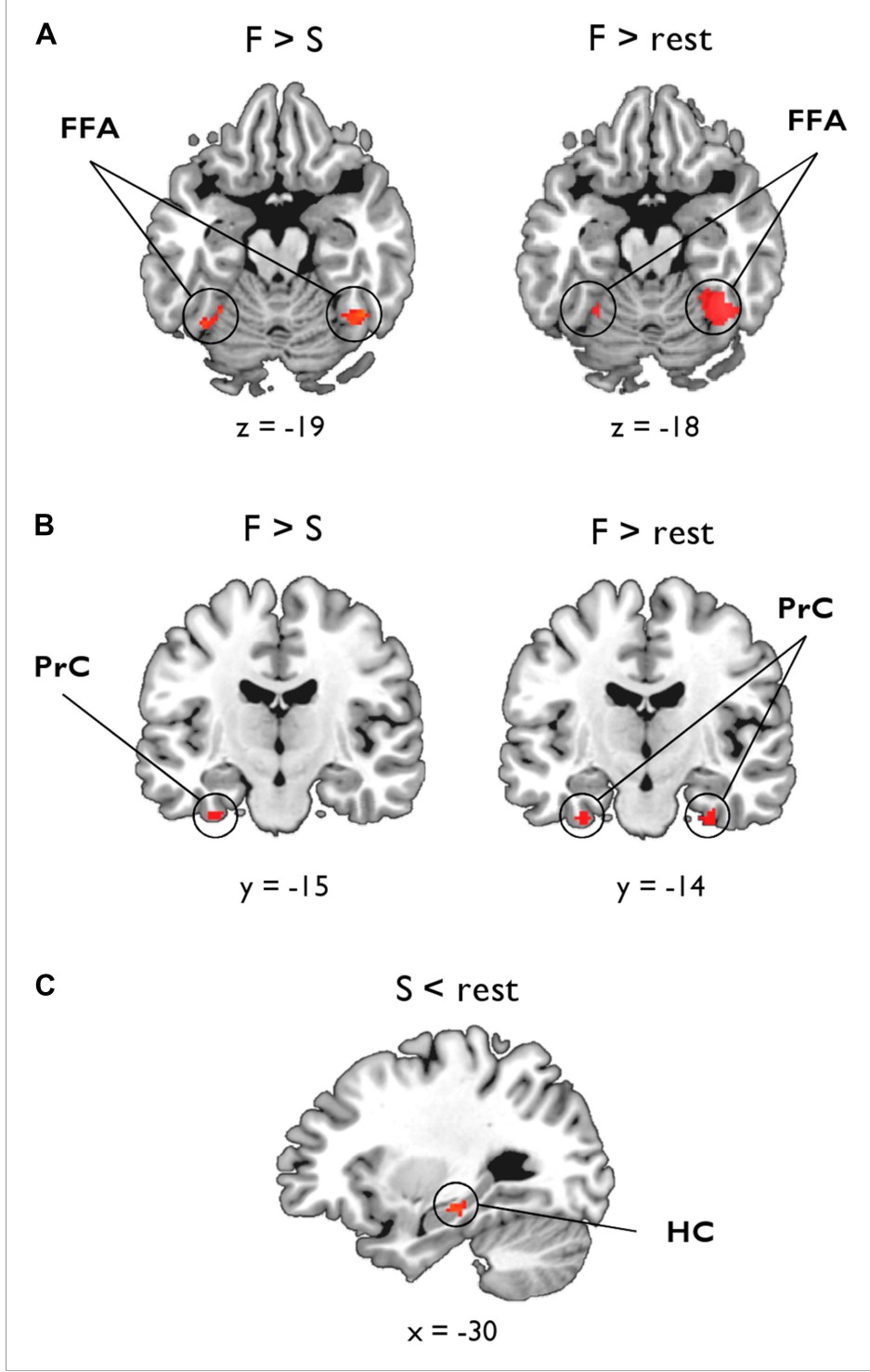

**Figure 3**. Voxel-wise linear associations between WM microstructure and category-sensitive blood oxygen level-dependent (BOLD) response. A group-level region-of-interest (ROI) analysis of the fMRI data was conducted to identify clusters reflecting a significant relationship between BOLD response for faces and scenes and tissue microstructure of the ILF and fornix, respectively. (**A**) Fusiform face area (FFA): significant bilateral clusters reflecting a negative association between BOLD response during face oddity judgements (F > S, left; F > rest, right) and ILF MD. (**B**) Perirhinal cortex (PrC): bilateral clusters reflecting a significant negative association between face-sensitive BOLD (F > rest) and ILF MD (right). A sub-threshold cluster for F > S is shown on the left. (**C**) Hippocampus (HC): a significant cluster was identified in the intermediate HC that corresponds to a positive association between task-induced scene deactivations (against rest) and fornix FA.

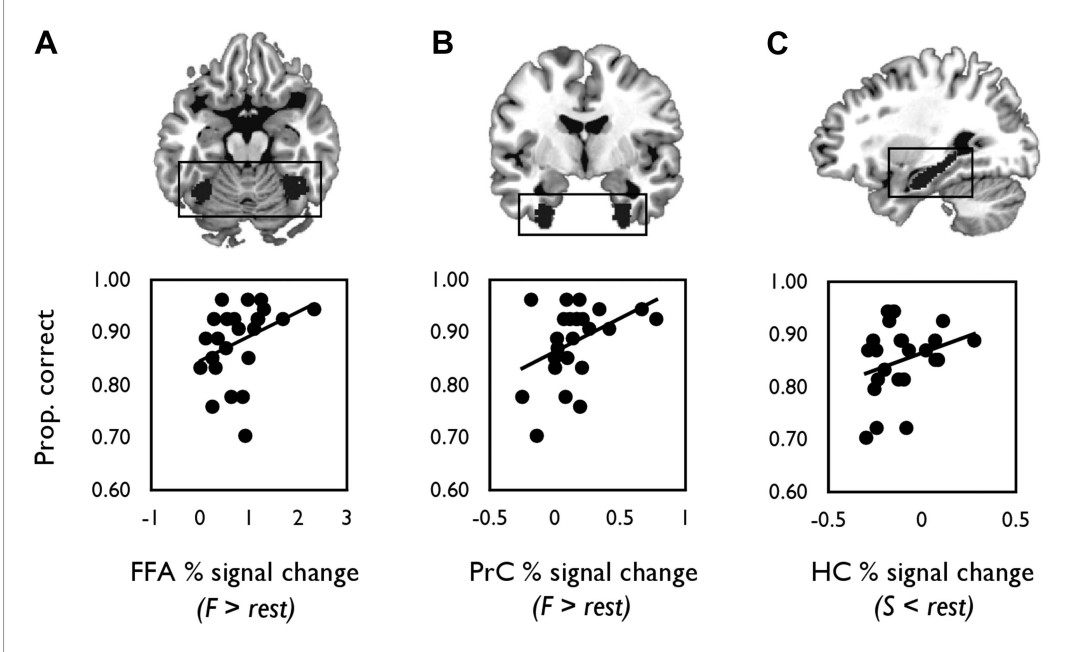

**Figure 4**. Correlations between mean percentage BOLD signal change from each probabilistic anatomical ROI (shown above each graph) and oddity performance. (**A**) Scatter plot displaying the relationship between inter-individual variation in percentage signal change for faces (relative to rest) and face oddity performance (proportion correct) in the pre-defined FFA ROI. (**B**) The relationship between inter-individual variation in face-related activations (against rest) and face oddity performance in the PrC ROI. (**C**) The relationship between task-induced scene deactivations (relative to rest) and scene oddity performance in the pre-defined HC ROI. A total of 24 data points are shown on each graph. Individual percentage signal change values for each ROI are contained in *Figure 4—source data 1*.

The following source data is available for figure 4:

**Source data 1**. Individual percentage BOLD signal change values for the fMRI contrasts.

## Mediation analysis: BOLD, DTI, and behaviour

From a mediation analysis conducted using ordinary least squares path analysis (*Preacher and Hayes, 2008*), FFA BOLD activity indirectly influenced face oddity performance through its relationship with ILF MD. As seen in *Figure 5A*, individuals with a higher FFA BOLD response for F > rest had significantly lower ILF MD values, and participants with lower ILF MD values showed significantly better face discrimination ability. A bootstrap 95% CI (based on 1000 bootstrapped samples) for the indirect effect was entirely above zero (see *Figure 5A*). Thus, there was no evidence that FFA BOLD activity influenced face oddity performance independent of its relationship with ILF MD. Further analyses revealed that this effect was predominantly evident in the right hemisphere (see *Supplementary file 2*). An alternative model in which FFA activity mediated the influence of ILF MD on face oddity performance revealed no evidence for an indirect effect of ILF MD on face discrimination accuracy through its effect on FFA BOLD activity (95% CI [−1017.19, 1361.64]; *Supplementary file 2*).

A comparable analysis for PrC revealed a significant direct effect of BOLD response for faces (F > rest) on face oddity success (*Figure 5B*). A bootstrap 95% CI for the indirect effect was found to cross zero and was thus non-significant (95% CI [−0.01, 0.1]). This finding indicates that PrC BOLD and ILF MD are independent predictors of face oddity performance. As with FFA, this effect was predominantly evident in the right hemisphere (see *Supplementary file 2*). The alternative mediation model in which PrC BOLD activity mediated the influence of ILF MD on face oddity (see *Supplementary file 2*) revealed no evidence for an indirect effect of ILF MD on face discrimination via its effect on PrC activity.

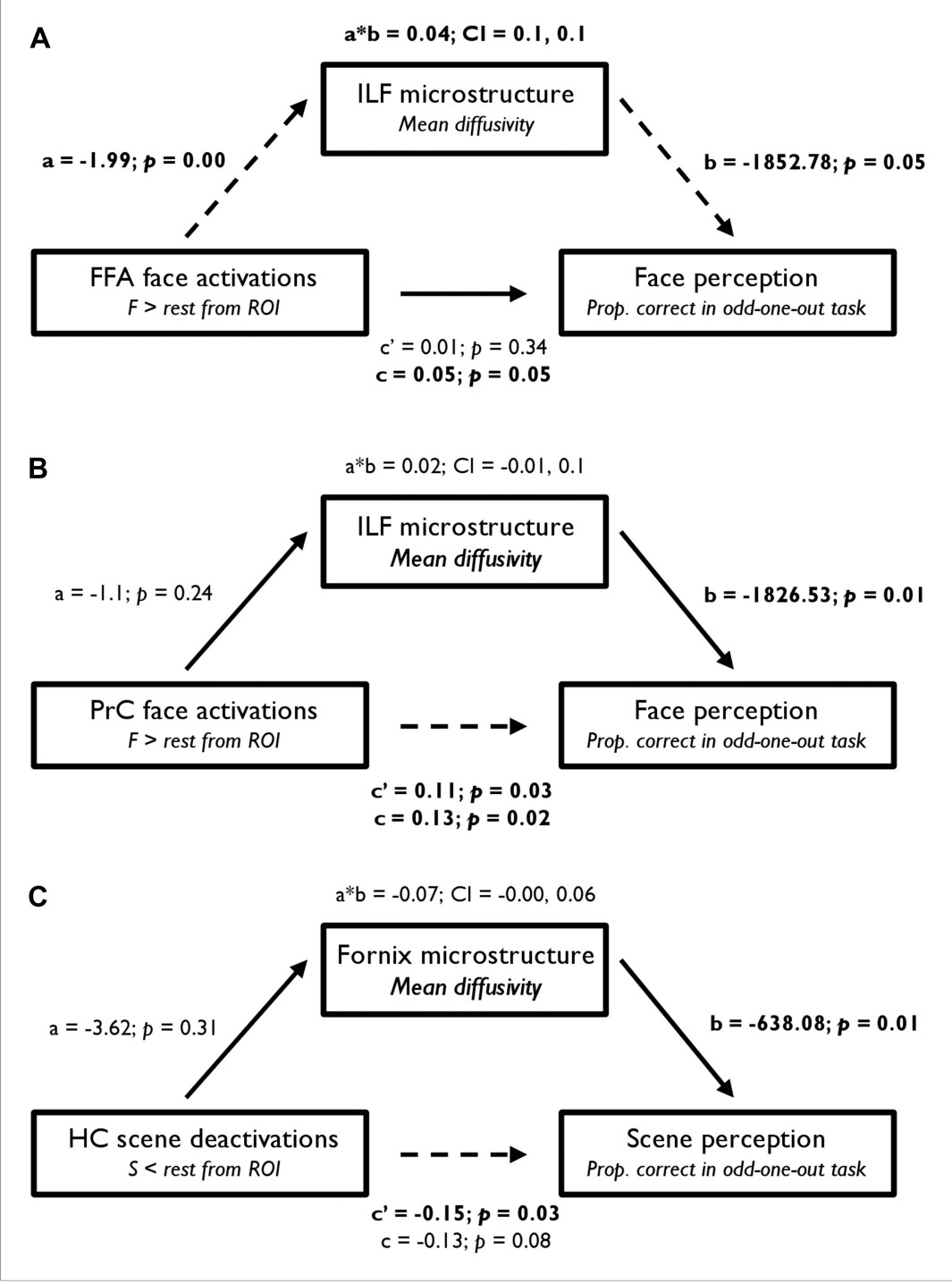

**Figure 5**. Statistical mediation (path) analysis examining the three-way relationship between regional functional BOLD activity, WM microstructure, and oddity performance. Mediation models are presented for (**A**) FFA, (**B**) PrC, and (**C**) HC. These models test the extent to which the relationship between BOLD response (in the a priori FFA, PrC, and HC ROIs) and odd-one-out accuracy is mediated by WM microstructure. The left, middle, and right boxes in each model represent the independent variable (IV), mediator (M), and dependent variable (DV), respectively. Unstandardised coefficients and their corresponding one-tailed p values are depicted for each path of interest (*a*, *b*, *a*b*, *c'*, and *c*). In a simple mediation model, these paths reflect the following: path *c* represents the total effect of the IV on the DV; path *a* quantifies the effect of the IV on the M; path *b* reflects the causal effect of the M on the DV; and path *c'* is the direct effect of the IV on the DV that also partials out the effect of the M. Significant paths are indicated by dashed lines and significant terms are indicated by bold font. The bootstrap 95% CI is displayed for the indirect effects (*a*b*). See **Supplementary file 2** for further details.

Analysis of the HC ROI revealed a significant direct effect of BOLD deactivation for S < rest on scene oddity (*Figure 5C*). As the indirect effect of HC BOLD, via fornix MD, crossed zero (−0.19, −0.01), these measures independently predicted inter-individual variations in scene oddity accuracy. The alternative model in which HC scene deactivations mediated the influence of fornix MD on scene discrimination performance (see *Supplementary file 2*) revealed no evidence for an indirect effect of fornix MD on scene oddity via its effect on HC BOLD.

## Discussion

Understanding network-level contributions to cognition is fundamental for modelling the relationship between brain structure and function and in elucidating the mechanisms underpinning inter-individual differences in behaviour. Here, we contribute new knowledge relevant to this goal by testing whether the distinct WM pathways converging on the PrC and HC, as measured by assessment of tissue microstructure in the ILF and the fornix, respectively, would be correlated with performance on face and scene odd-one-out discriminations that are differentially sensitive to PrC and HC damage in humans (*Lee et al., 2005a*) and monkeys/rats (*Buckley et al., 2001*; *Bartko et al., 2007*). We found that MD of the fornix, a principal tract linking the HC with surrounding cortical and subcortical areas (*Saunders and Aggleton, 2007*; *Aggleton et al., 2015*), was strongly (negatively) associated with perceptual performance on scene, but not face, oddity judgements (*Figure 1*). Conversely, MD for the ILF, the main ventral visual input pathway to antero-medial temporal cortex (*Catani et al., 2003*; *Latini, 2015*), correlated with face, but not scene, oddity performance. None of the microstructural measures obtained, in either pathway, were significantly associated with accuracy on a difficulty-matched size oddity condition, consistent with the preservation of size oddity after MTL lesions (*Buckley et al., 2001*).

Going beyond this striking double dissociation, we also demonstrated selective relationships between WM microstructure and the magnitude of BOLD response in key, category-selective, ROIs situated along these tracts (*Figure 3*). Specifically, ILF MD was strongly associated with BOLD response for faces in both PrC and FFA—two regions linked anatomically by the ILF (*Pyles et al., 2013*). In turn, average percentage signal change from our predefined FFA and PrC ROIs was related to face discrimination success. Notably, WM microstructure (MD) of the ILF mediated the relationship between BOLD activity in FFA and oddity performance for faces; this was not evident for PrC.

By contrast, fornix FA was positively associated with HC scene deactivations. Furthermore, inter-individual differences in HC BOLD percentage signal reduction, within our independently defined HC ROI, were related to performance in the scene oddity task. A mediation analysis revealed that fornix WM and HC BOLD deactivations made independent contributions to scene discrimination performance.

Our data, therefore, make a novel and important contribution to the literature by demonstrating that inter-individual differences in fornix and ILF microstructure may play an integral role in determining performance on complex visual discriminations for different visual categories. Further, while previous studies in both animals and humans have linked fornix connectivity with both spatial learning (*O'Keefe et al., 1975*; *Buckley et al., 2001*; *Hofstetter et al., 2013*; *Dumont et al., 2015*) and memory (*Gaffan, 1994*; *Rudebeck et al., 2009*; *Vann et al., 2009*; *Bennett et al., 2014*), this is the first demonstration that inter-individual variability in this pathway in humans is associated with spatial scene perception. The striking convergence between the findings from our DTI analyses, and similar functional divisions evident in complementary fMRI and animal/human MTL lesion studies (e.g., *Buckley et al., 2001*; *Lee, Bussey, et al., 2005*), provides strong evidence for HC and PrC contributions to perceptual discriminations involving complex visual stimuli, via their role as key nodes within distinct, distributed, functionally specialised neural networks.

According to hierarchical representational accounts, PrC is best understood as the apex of the visual ventral processing stream (*Murray et al., 2007*). Several regions along this stream have been identified as important for face processing (*Tsao et al., 2008*), including FFA (*Kanwisher et al., 1997*), but also more recently PrC, which seems to play a critical role in face perception via its contribution to processing of complex feature conjunctions (*Barense et al., 2005*, *2007*). For example, monkeys with PrC lesions are impaired on oddity tasks when presented with face and object arrays (*Buckley et al., 2001*); this profile is also seen in patients when damage to the MTL includes PrC (*Lee et al., 2005a*). By using a similar oddity task here (*Barense et al., 2005*; *Lee et al., 2005a*), our results indicate that the ILF, through its interactions with more posterior face processing regions (e.g., FFA), may be important in facilitating the formation of complex representations 'downstream' in the MTL. This extends previous

findings that highlight a functional coupling between anterior temporal lobe regions, such as PrC, and face processing regions of fusiform cortex (*Moeller et al., 2008*; *O'Neil et al., 2012*, *2013*, *2014*; *Anzellotti et al., 2014*). The results presented here also go beyond such findings by presenting a 'structural realisation' of this functional connectivity (*Kosslyn and Van Kleeck, 1990*; *Behrmann and Plaut, 2013*), that is, a direct relationship between the WM bundle connecting these distributed regions and complex face discrimination. Moreover, we demonstrate that ILF tissue microstructure is associated with face-related BOLD activity in both PrC and FFA. Together, these analyses provide clear links between structure, function, and behaviour, and support the idea that the anatomical connection linking antero-medial temporal cortex (including PrC) and FFA is a critical structure in complex face perception, as suggested by neuropsychological studies (*Thomas et al., 2009*; *Grossi et al., 2014*).

An important question that emerges from these results is how inter-individual variation in behaviour emerges from the interplay between ILF microstructure and functional activity in the face-processing network. Studies exploring the role of functional nodes in antero-medial temporal and occipital temporal cortex suggest a number of possible mechanisms for how these regions might *together* support face processing. For instance, both antero-medial temporal cortex (*Barense et al., 2010*; *Freiwald and Tsao, 2010*; *Collins and Olson, 2014*; *Yang et al., 2014*; *Anzellotti and Caramazza, 2015*) and ventral occipital temporal cortex (*Winston et al., 2004*; *Nestor et al., 2011*; *Anzellotti et al., 2014*) have been shown to contain representations that are invariant to facial transformations (e.g., viewpoint or emotional expression), indicating that these discrete regions may be involved jointly in the online maintenance of viewpoint-invariant face representations (*Freiwald and Tsao, 2010*; *Nestor et al., 2011*; *Collins and Olson, 2014*). Another study that manipulated the visual similarity between face stimuli found that PrC exhibits a greater response to highly overlapping faces, whereas FFA shows the opposite pattern (*Mundy et al., 2012*), a finding consistent with a posterior-to-anterior hierarchy in which face representations become increasingly complex (*Saksida and Bussey, 2010*). Furthermore, a recent study reported response suppression for different images of matching identities in antero-medial temporal cortex but not more posterior face-processing regions (*Yang et al., 2014*). Strikingly, this adaptation effect in anterior temporal lobe was preserved in a prosopagnosic patient with ipsilateral lesions of FFA and 'occipital face area', indicating a potential top–down role of this region in processing face identity. By observing a specific relationship between the ILF and performance on our oddity task in which highly overlapping faces must be discriminated across multiple viewpoints, our results suggest that the connection established by the ILF may be integral for both (a) an effective iterative feedback mechanism between PrC and ventral occipital temporal cortex that allows for online maintenance of identity across visual transformations (*Fox et al., 2008*; *Freiwald and Tsao, 2010*; *Nestor et al., 2011*; *O'Neil et al., 2013*; *Yang et al., 2014*) and (b) the efficient feed forward of stimulus information from fusiform gyrus and extrastriate cortex to antero-medial temporal cortex, which permits fine-grained discrimination across multiple viewpoints (*Graham et al., 2010*; *Saksida and Bussey, 2010*; *Fox et al., 2013*). This proposed relationship is confirmed by evidence of a strong coupling between ILF MD and face-sensitivity in both FFA and PrC, as well as recent evidence suggesting that patterns of WM connectivity are better predictors of an individual's FFA location than group-derived functional ROIs (*Saygin et al., 2012*; *Osher et al., 2015*). These results may also address some inconsistencies in the literature, where associations have been found between face recognition and locally defined ventral temporal pathways but not with long-range ILF (*Tavor et al., 2014*; *Gomez et al., 2015*). As indicated above, it may be the case that tasks probing complex antero-medial temporal representations (e.g., perceptual identity) will reveal stronger associations with long-range ILF connectivity (see also *Postans et al., 2014*).

The importance of WM in driving cognitive performance was made explicit by the finding that ILF microstructure mediated the relationship between BOLD activity in FFA and accuracy on face discrimination. This result converges with data from individuals with CP, in whom the FFA functions normally, but who show disrupted ILF microstructure and macrostructure and a reduction in posterior–anterior temporal lobe connectivity under both task-related and resting conditions (*Thomas et al., 2009*; *Avidan et al., 2014*). Notably, ILF undergoes protracted development well into adulthood, and these developmental changes in ILF MD are tightly and specifically linked with an age-related increase in the size of the FFA (*Scherf et al., 2014*). Our findings, together with the refinement of the ILF over an extended developmental period, and its compromise in CP, all point to a potential mechanism in which an extended face network, over the course of experience and maturation, becomes progressively organised and optimised. This may occur via neural activity-dependent

mechanisms that can stimulate myelination or myelin remodelling, thereby leading to increased network specialisation (*Scherf et al., 2014*; *McKenzie et al., 2014*).

Involvement of the ILF in face perception was further confirmed by a complementary whole-brain TBSS analysis. This augmented our deterministic approach by highlighting additional associations between face oddity performance and microstructural variation in the WM tracts linking occipital and temporal lobe structures (including PrC) with the frontal lobes (e.g., IFOF, SLF, and the cingulum bundle [*Yeterian and Pandya, 2010*]). These WM pathways may be necessary for linking perceptual processing of faces in occipital cortex and vATL with prefrontal cortex face representations (*Moeller et al., 2008*; *Tsao et al., 2008*), again highlighting the critical nature of broadly distributed circuits in face perception.

By contrast, inter-individual differences in fornix microstructure were associated with performance on a complex scene discrimination task and scene-related BOLD deactivations in HC; these findings support the notion that the HC—as part of a broader anatomical network of which the fornix is a key component—is involved in spatial processing (*Aggleton et al., 2015*). While the importance of the HC in spatial navigation has long been established at the neurophysiological level (*Ono et al., 1991*; *O'Keefe et al., 1998*; *Rolls, 1999*), recent studies have since indicated that the HC is behaviourally important when tasks place a demand on complex spatial representations (*Bird and Burgess, 2008*; *Graham et al., 2010*; *Mundy et al., 2012*). With this in hand, therefore, it is worth considering exactly how the fornix, as a pathway between distributed regions, contributes to spatial scene perception and the complex spatial representations contained in HC.

In particular, the reciprocal interplay between HC and surrounding neocortical and subcortical regions (*Saunders and Aggleton, 2007*; *Aggleton et al., 2015*)—that is afforded partly by fornical connections—appears critical for the formation of flexible spatial representations in the HC (i.e., those that maintain the coherent layout of a spatial environment across multiple viewpoints). For example, efferent connections from the HC to both the mammillary bodies and the anterior thalamus, via the fornix, have been shown to play a role in scene processing (*Gaffan et al., 2001*) and object-in-place learning (*Gaffan, 1994*; *Parker and Gaffan, 1997*; *Buckley et al., 2004*). Fornix lesions also cause object-in-place learning impairments above and beyond combined lesions to frontal and inferior temporal (i.e., ventral stream) regions (*Wilson et al., 2008*). This suggests that visual-spatial inputs from dorsal visual areas (e.g., parahippocampal and posterior cingulate cortices), via the subiculum of the HC, may underpin aspects of scene processing that are independent of interactions between inferotemporal and frontal cortices. These dorsally mediated inputs may, for example, convey spatial, rather than object, feature information, such as orientation, position, and size (*Buckley et al., 2004*; *Wilson et al., 2008*; *Nasr et al., 2014*). Interestingly, this may also account for the moderate, though non-significant, association between size oddity and fornix microstructure (see 'Results').

Given the effect of fornix lesions on these various forms of spatial processing, it is plausible that tasks that tap these emergent flexible scene representations, either by the use of different viewpoints (as in oddity tasks) or where there is a need to discriminate between scenes or objects with unique conjunctions of spatial features (*Buckley et al., 2004*), may be particularly sensitive to the HC and the functional network it forms via the fornix. Consistent with this, patients with HC damage only show scene oddity impairments when items are presented from different viewpoints (*Lee et al., 2005b*), and individuals with Alzheimer's disease exhibit greater deficits on scene odd-one-out tasks when different, rather than same, viewpoint scenes are presented (*Lee et al., 2006*). Likewise, HC damage leads to short-term memory deficits in matching rotated scenes based on topographical information (*Bird and Burgess, 2008*). As representational accounts also propose that episodic retrieval is predominantly driven by reimagining the rich spatial context in which a particular memory event occurred (*Gaffan, 1994*; *Hassabis and Maguire, 2007*; *Graham et al., 2010*), this may explain why individual differences in fornix microstructure have also been associated with non-spatial, episodic memory tasks (*Metzler-Baddeley et al., 2011*).

Interestingly, the reported association between fornix microstructure (FA, and to a lesser extent MD) and scene-related BOLD activity in HC was in the opposite direction to that observed between ILF microstructure and face selectivity in PrC/FFA, with fornix FA positively correlating with HC scene *deactivations*. Further, this surprising association was localised in more anterior/intermediate HC (*Figure 3C*). In contrast, posterior HC is more often recruited during visual perception (i.e., oddity; *Lee et al., 2005a*; *Mundy et al., 2013*; *Zeidman et al., 2014*). Given differences in subfield organisation (*Duvernoy, 1988*; *Poppenk et al., 2013*) and fornical axon fibre contributions

(*Saunders and Aggleton, 2007*; *Aggleton, 2012*) along the long axis, it is likely that the more anterior HC plays a different functional role compared to posterior HC (*Nadel et al., 2013*; *Poppenk et al., 2013*; *Duarte et al., 2014*; *Zeidman et al., 2014*). Notably, using a similar scene oddity paradigm, *Lee et al. (2008)* reported significant scene activations in posterior HC but scene *deactivations* (relative to baseline) in anterior HC. Negative HC BOLD response has also been localised to anterior rather than posterior HC during spatial encoding and retrieval, consistent with the results reported here (*Figure 3C*; *Duarte et al., 2014*).

This interpretation of scene deactivations (relative to baseline) should be treated with caution given that baselines are difficult to define in functional neuroimaging studies (*Gusnard and Raichle, 2001*), particularly in MTL regions such as the HC (*Stark and Squire, 2001*). For instance, it is possible that the reported association emerges from variability in anterior HC activity during rest, rather than during scene oddity judgements. Further, not only is it difficult to define a baseline in HC, but there is also a particularly complex relationship between HC neural activity and the BOLD response (*Ekstrom, 2010*). Studies in both rats (*Schridde et al., 2008*; *Angenstein et al., 2009*) and humans (*Ekstrom et al., 2009*) have reported increases in neural activity that are associated with negative changes in the HC BOLD signal. The BOLD signal itself is dependent on the relationship between cerebral blood flow and oxygen metabolism (*Fox and Raichle, 1986*), and the assumption that, during neural activity, blood flow outmatches metabolic demands (leading to a relative increase in oxygenated haemoglobin). One possibility, therefore, is that sparser blood supply in the HC (e.g., lower capillary density; *Borowsky and Collins, 1989*) leads to a decoupling between neuronal activity and BOLD, that is, where oxygen metabolism exceeds local blood flow.

Whilst HC BOLD deactivations were related to both fornix microstructure and scene discrimination accuracy, fornix microstructure did not mediate the relationship between HC BOLD and behavioural performance (and vice versa). More specifically, HC activity and fornix microstructure independently contribute to individual variability in scene discrimination performance. Further, in the BOLD-DTI analyses, fornix microstructure was only found to correlate with HC BOLD (*Figure 3C*) when the contrast was between scenes and rest, not scene and faces. The distinct contributions of HC neuronal activity and fornix microstructure are consistent with several lines of evidence from lesion work in rats. First, that non-fornical HC pathways are also critical for spatial processing (*Dumont et al., 2015*). Second, that fornix lesions, which impair HC-dependent spatial memory, do not necessarily suppress HC neuronal activity but rather disrupt longer term HC cellular plasticity mechanisms (*Fletcher et al., 2006*). Third, that fornical fibres may mediate some spatial functions not attributable to the HC (*Whishaw and Jarrard, 1995*). In summary, our findings confirm that scene processing (and by extension episodic memory) is an emergent property of the functional and structural connectivity between the HC and key cortical and subcortical regions (*Graham et al., 2010*), mediated in part, but not exclusively, by the fornix.

It is notable that, overall, we observed stronger statistical effects with MD compared to FA, although FA did show a similar profile to that seen in MD (e.g., for fornix FA and scene oddity), and was associated with a cluster in ILF for face oddity in our whole brain TBSS analysis. As different attributes of WM (e.g., axon density, axon diameter, myelin [*Beaulieu, 2002*], and the manner in which axons are laid out within a given voxel [*Jones et al., 2013*]) can influence the hindrance and restriction of water, as has been described elsewhere (*Jones et al., 2013*), the interpretation of DTI and its specific metrics (including MD and FA) is not straightforward. Thus, while we found stronger associations with MD than with FA, we are not yet able to say whether a particular aspect of WM microstructure (e.g., myelin) underpins these differences. Consistent with this, MD and FA are often reported jointly in the literature (*Metzler-Baddeley et al., 2011*; *Gschwind et al., 2012*; *Scherf et al., 2014*), and like our reported findings, sometimes MD has been shown to have stronger effects than FA. For example, MD can decrease following spatial learning (*Sagi et al., 2012*) and appears to be more sensitive than FA to age-related changes in ILF and fornix WM (*Scherf et al., 2014*; *Wendelken et al., 2014*). Further, MD and FA metrics are not orthogonal, meaning that changes in one of these measures will be potentially reflected in the other (*O'Donnell and Pasternak, 2015*).

Based on representational accounts of MTL function that assume dissociable roles for the HC and PrC in scene and face processing, respectively (*Graham et al., 2010*; *Saksida and Bussey, 2010*), we provide a compelling demonstration that WM tracts connecting to the HC and PRC may be critical pathways in networks that support the successful discrimination of complex places and faces, respectively. More specifically, complex face perception is not just a property of the FFA or PrC but

emerges from interactions within large-scale integrated neurocognitive networks (*Mesulam, 1990*, *1994*; *Behrmann and Plaut, 2013*). Likewise, spatial impairments in amnesic individuals may not simply disrupt local HC processing per se, but rather the extent to which communication in the broader, functional networks supporting formation of flexible spatial representations is interrupted (*Graham et al., 2010*; *Murray and Wise, 2010*; *Baxter, 2012*). By providing a structural framework that may underpin how category-selective perception emerges in the MTL, these findings add to an emerging literature that challenges the long-held view that the MTL is an exclusive unitary memory system. Rather, our results indicate that these MTL substructures, through their distinct anatomical connections (including fornix and ILF), comprise broader neurocognitive networks that are dissociable in the types of stimulus representations they support. In this context, higher level perception, rather than depending on the isolable properties of individual MTL regions, arises from the dynamic interplay within integrated and specialised neurocognitive circuits (*Gaffan, 2002*; *Graham et al., 2010*; *Saksida and Bussey, 2010*).

## Materials and methods

### Participants

30 undergraduates from Cardiff University participated in this study (2 male; aged 18–22 years; mean = 19; SD = 0.96) and were paid for taking part. The experiment was undertaken with the understanding and written consent of each subject. Cardiff University School of Psychology Research Ethics Committee approved the research project.

### Oddity task and procedure

In the oddity task, participants were presented with three stimuli on each trial (top centre; bottom left; bottom right) and instructed to select the odd-one-out as quickly and as accurately as possible. Two of these stimuli were the same item from different viewpoints, and the third stimulus was a different item. In this article, we analysed the behavioural performance on scene and face oddity with a size oddity condition acting as a single feature baseline. Example trials for the scene and face oddity conditions are shown in *Figure 1*.

The scene stimuli were real-world photographs of outdoor environments. On each trial, participants viewed two images of a single locale from different viewpoints and one different locale. Face stimuli were greyscale photographs of human faces, half of which were male. Individual faces were overlaid on a black background (170 × 216 pixels). Two faces were the same individual presented from different viewpoints, and the target was a different face presented from a different viewpoint. For the size task, three black squares were presented. The position of the squares on the screen was jittered so that none of the edges lined up along vertical or horizontal axes. On each trial, two of the squares were identical in size and a third square was either slightly larger or smaller. The difference in length between target and non-targets could vary between 9 and 15 pixels. All stimuli were trial-unique (i.e., never repeated once shown in the task).

Each trial was presented for 6 s with a jittered inter-trial interval of 500–4000 ms. The task was administered in the scanner over three functional imaging runs. Within each run, trials for a given category (scene, face, size) were presented in mini-blocks of three successive trials. The order in which category 'triplets' were presented was counterbalanced across participants. Overall, 18 trials were presented per category per run resulting in 54 trials per condition overall. An equal number of targets appeared at each screen position (i.e., top centre; bottom left; bottom right) within each stimulus condition. Stimuli were presented in the scanner using ePrime (Psychology Software Tools, Inc., Sharpsburg, PA) and projected onto the screen behind the participant using a Canon SX60 LCOS projector system combined with the Navitar SST300 zoom converter lens. Button responses in the scanner were acquired using a right-hand MR compatible button box.

### MRI data acquisition

Imaging data were collected at the Cardiff University Brain Research Imaging Centre (CUBRIC) using a GE 3-T HDx MRI system with an 8-channel receive-only head coil. Whole-brain high angular resolution diffusion image data were acquired using a diffusion weighted single-shot spin-echo echo-planar imaging (EPI) pulse sequence with the following parameters: TE = 87 ms;

voxel dimensions = 2.4 × 2.4 × 2.4 mm³; field of view = 23 × 23 cm²; 96 × 96 acquisition matrix; 60 contiguous slices acquired along an oblique–axial plane with 2.4-mm thickness (no gap). Acquisitions were cardiac gated using a peripheral pulse oximeter. Gradients were applied along 30 isotropic directions (*Jones et al., 1999*) with b = 1200 s/mm². Three non-diffusion-weighted images were acquired with b = 0 s/mm². Functional BOLD data were acquired using an EPI pulse sequence with the following scan parameters: TR/TE = 3000/35 ms; FOV = 240 mm; 64 × 64 data matrix; ASSET (acceleration factor); 90° flip angle. 42 interleaved slices were collected per volume for whole-brain coverage. Each slice was 2.8-mm thick with a 1-mm inter-slice gap (3.75 × 3.75 × 2.8-mm voxels). Slices were acquired with a 30° axial-to-coronal tilt relative to the AC-PC line (anterior upwards). The first four volumes of each run were discarded to allow for signal equilibrium. Two 3D SPGR images were acquired to improve registration and reduce image distortion as a result of magnetic field inhomogeneity (TE = 7 ms and 9 ms, TR = 20 ms, FOV = 384 × 192 × 210 mm, 128 × 64 × 70 data matrix, 10° flip angle). The SPGR used the same slice orientation as the functional acquisition. High-resolution anatomical images were acquired using a standard T1-weighted 3D FSPGR sequence comprising 178 axial slices (TR/TE = 7.8/3.0 s, FOV = 256 × 256 × 176 mm, 256 × 256 × 176 data matrix, 20° flip angle, and 1 mm isotropic resolution).

## Diffusion MRI pre-processing

ExploreDTI (*Leemans and Jones, 2009*) was used to correct for subject motion and eddy current distortions. In order to correct for partial volume artefacts arising from voxel-wise free water contamination, the two-compartment 'free water elimination' procedure was implemented (*Pasternak et al., 2009*). Following free water correction, corrected diffusion indices were computed: MD and FA. The resulting free water-corrected maps were inputs for both the tractography and the TBSS analyses.

## Tractography

Deterministic whole-brain WM tractography was performed using ExploreDTI. Tractography was based on constrained spherical deconvolution (CSD; see *Tournier et al., 2004*; *Jeurissen et al., 2011*), which extracts peaks in the fibre orientation density function (fODF) at each voxel. The 'diffusion tensor' model is not sufficient when modelling the distribution of water displacement in more complex fibre configurations, such as crossing or kissing fibres (e.g., as seen where the anterior pillars of the fornix meet the anterior commissure). The fODF—which is estimated directly by CSD—quantifies the proportion of fibres in a voxel pointing in each direction and so information about more complex fibre configurations can be extracted (*Jones, 2008*). Each streamline was reconstructed using an fODF amplitude threshold of 0.1 and a step size of 1 mm and followed the peak in the fODF that resulted in the smallest step-wise change in orientation. An angle threshold of 30° was used and any streamlines exceeding this threshold were terminated.

To generate three-dimensional reconstructions of each tract, 'way-point' ROIs were manually drawn onto whole-brain FA maps in the diffusion space of individual subjects (*Metzler-Baddeley et al., 2011*). In accordance with Boolean logic, these way-point ROIs can specify that: (a) tracts passing through multiple ROIs are retained for analysis (i.e., 'AND' ROIs) and (b) tracts passing through certain ROIs are omitted from analysis (i.e., 'NOT' ROIs). Depending on the specific tract, or the anatomical plausibility of initial reconstructions, such ROIs can be combined; for example, a tract may pass through ROI-1 'AND' ROI-2 but 'NOT' ROI-3 (*Figure 6*). The ROI approaches described below will adopt this Boolean terminology when describing the ROIs that were drawn for each tract. Following the reconstruction of each pathway in each subject, mean MD and FA were calculated by averaging the individual values at each 1-mm step along the tracts, and in the case of the ILF, across hemispheres. The placement of ROIs for each tract is depicted in *Figure 6*.

### Fornix ROIs

A multiple ROI approach was used to reconstruct the fornix (*Metzler-Baddeley et al., 2011*). The approach involved placing a seed point ROI on the coronal plane where the anterior pillars enter the main body of the fornix. A single AND ROI is then positioned on the axial plane, encompassing both crus fornici at the lower part of the splenium of the corpus callosum. Three NOT ROIs are then placed: (1) anterior to the fornix pillars, (2) posterior to the crus fornici, and (3) on the axial plane, intersecting the corpus callosum. Once these ROIs were placed, and the tracts reconstructed, anatomically implausible fibres were removed using additional NOT ROIs.

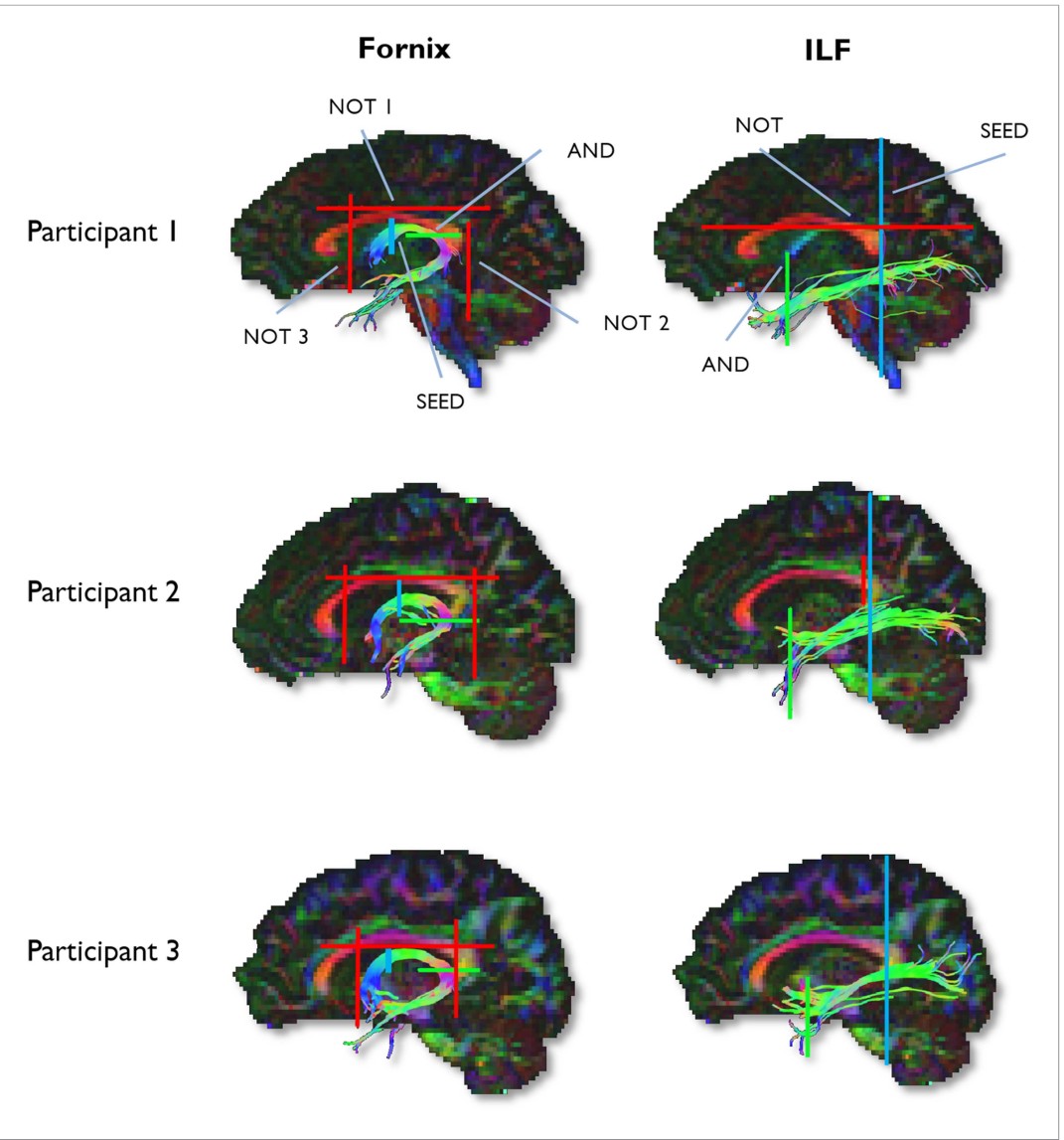

**Figure 6**. Example reconstructions for the fornix and ILF. Tractography ROIs are shown for three participants (SEED ROI, blue; AND ROI, green; NOT ROI, red).

## ILF ROIs

Fibre-tracking of the ILF was performed using a two-ROI approach in each hemisphere (*Wakana et al., 2007*). First, the posterior edge of the cingulum bundle was identified on the sagittal plane. Reverting to a coronal plane at this position, a SEED ROI was placed that encompassed the whole hemisphere. To isolate streamlines extending towards anterior temporal cortex, a second ROI was drawn at the most posterior coronal slice in which the temporal lobe was not connected to the frontal lobe. Here, an additional AND ROI was drawn around the entire temporal lobe. Like the fornix, any outlying streamlines were removed using additional NOT ROIs. This approach was carried out in both hemispheres; microstructural measurements for the left and right ILF (for both MD and FA) were averaged to provide a bilateral estimate of ILF microstructure.

## TBSS

Voxel-wise statistical analysis of the DTI data was carried out using TBSS (for a full description, see *Smith et al., 2006*). TBSS permits voxel-wise correlations by non-linearly projecting all subjects' DTI

data (MD or FA) onto a mean tract skeleton before applying voxel-wise cross-subject statistics. To investigate the relationship between microstructure (MD, FA) and performance on the behavioural tasks, we implemented a GLM with two explanatory variables: proportion correct for (1) scene oddity and (2) face oddity. Two main contrasts were carried out to compare our two categories of interest: Faces > Scenes (F > S), and Scenes > Faces (S > F). To address the possibility of reporting false-positives, clusters were extracted using Threshold-Free Cluster Enhancement (*Smith and Nichols, 2009*) with a FWE-corrected threshold of p = 0.05. All reported co-ordinates are in Montreal Neurological Institute (MNI) 152 space.

## fMRI pre-processing

Functional MRI data were preprocessed using FEAT (FSL, www.fmrib.ox.ac.uk/fsl) and involved: motion correction (*Jenkinson et al., 2002*), removal of non-brain tissue using BET (*Smith, 2002*), spatial smoothing using a 5 mm full-width at half-maximum (FWHM) Gaussian kernel, mean-based intensity normalisation, and high-pass temporal filtering (Gaussian-weighted least squares straight line fitting, with sigma = 50.0 s). Phase information from the two SPGR images was unwrapped using PRELUDE (*Jenkinson, 2003*). These phase images were then subtracted and the resulting fieldmap used to unwarp the EPI data using FUGUE. Time-series statistical analysis was carried out using FMRIB's Improved Linear Model (FILM) with local autocorrelation correction (*Woolrich et al., 2001*). Registration to high-resolution 3D anatomical T1 scans (per participant) and to a standard MNI 152 template image (for group averaging) was carried out using FLIRT (*Jenkinson and Smith, 2001*; *Jenkinson et al., 2002*). Following pre-processing, analyses were first conducted at the single-subject level using FILM. The BOLD signal was modelled using a standard model of haemodynamic response function. Co-ordinates of significant effects are reported in MNI space.

## fMRI data analysis

Two explanatory variables comprising correct scene and face oddity trials were used to model the time-course data. A GLM was implemented to examine the BOLD response associated with three main contrast groups: (a) each separate stimulus category compared to each other (S > F; F > S), (b) each stimulus category against rest baseline (S > rest; F > rest), and (c) rest baseline against scenes and faces (i.e., negative BOLD activation for scenes and faces). The three individual runs for each participant were combined using a fixed-effects model in FEAT. Group-level ROI analyses were carried out using the FMRIB Local Analysis of Mixed Effects tool (*Beckmann et al., 2003*; *Woolrich et al., 2004*). A bilateral ROI of the HC was extracted from the Harvard–Oxford subcortical atlas (*Jovicich et al., 2009*) using FSL and was defined using a probability threshold of 50%. The PrC was based on a probabilistic mask from the literature (*Devlin and Price, 2007*) and likewise defined using a probability threshold of 50%. The FFA was approximated by intersecting an anatomical mask of the fusiform gyrus (from the Harvard–Oxford cortical atlas) with a probabilistic map of face activations thresholded at 50% (Atlas of Social Agent Perception, http://www.andrewengell.com/wp; see *Engell and McCarthy, 2013*). Anatomically defined, independent ROIs were also used for the DTI-BOLD analysis of other scene-selective regions. For the PHG, we used posterior PHG from the Harvard–Oxford Cortical Atlas, defined using a probability threshold of 50%. For RSC, we extracted Brodmann area 29 dilated by a single voxel (*Bluhm et al., 2009*). TOS was taken from the International Consortium for Brain Mapping (ICBM) Sulcal atlas using a probability threshold of 25% (*Mazziotta et al., 1995*).

DTI metrics (MD and FA) for the fornix and ILF were demeaned and added as covariates for the HC and PrC/FFA group-level analyses, respectively. The resulting statistical maps were thresholded at p = 0.01 (voxel-wise, uncorrected), where supra-threshold voxels reflect a significant positive association between BOLD response for a given contrast (see above) and a particular microstructural measure. To determine *negative* associations with MD, individuals' MD value was multiplied by minus 1 to derive inverse values. To control for false positives in each of the ROIs, we used Monte-Carlo simulation (AFNI's 3dClustSim, http://afni.nimh.nih.gov/pub/dist/doc/program_help/3dClustSim.html) to determine cluster significance at our voxel-wise alpha level (p = 0.01, see above); all reported clusters correspond to cluster-corrected threshold of p < 0.01. To correlate BOLD response and behaviour, percentage signal change values for the four main contrasts (S > F, F > S, S > rest, F > rest) were extracted from the ROIs using Featquery in FSL. These percentage signal change values (extracted from our main probabilistic anatomical ROIs) were also used in the mediation analyses.

## Acknowledgements

This work was supported by the Medical Research Council (G1002149), Biotechnology and Biological Sciences Research Council (BB/I007091/1), and the Waterloo Foundation. Derek K Jones was supported by a Wellcome Trust New Investigator Award. We would like to thank Carina Hibbs and Mia Schmidt-Hansen for help with task design, John Aggleton for advice about fornix connectivity, and Nils Mulhert and Katja Umla-Runge for valuable discussion. We are especially grateful to Ofer Pasternak and Greg Parker for providing the free water correction pipeline.

## Additional information

### Funding

| Funder | Grant reference | Author |
| --- | --- | --- |
| Medical Research Council (MRC) | G1002149 | Carl J Hodgetts, Kim S Graham |
| Biotechnology and Biological Sciences Research Council (BBSRC) | BB/I007091/1 | Mark Postans, Kim S Graham |
| Waterloo Foundation | | Kim S Graham |
| Wellcome Trust | New Investigator Award | Derek K Jones |
| Cardiff University | School of Psychology PhD Studentship | Jonathan P Shine |

The funders had no role in study design, data collection and interpretation, or the decision to submit the work for publication.

### Author contributions

CJH, Conception and design, Acquisition of data, Analysis and interpretation of data, Drafting or revising the article; MP, JPS, DKJ, Analysis and interpretation of data, Drafting or revising the article; ADL, KSG, Conception and design, Analysis and interpretation of data, Drafting or revising the article

### Author ORCIDs

Carl J Hodgetts, http://orcid.org/0000-0002-0339-2447
Andrew D Lawrence, http://orcid.org/0000-0001-6705-2110
Kim S Graham, http://orcid.org/0000-0002-1512-7667

### Ethics

Human subjects: The study was approved by the School of Psychology, Cardiff University Ethics Committee. Written informed consent was obtained from each participant before taking part in the experiment, including consent to publish results.

## Additional files

### Supplementary files

• Supplementary file 1. Table of co-ordinates for the whole-brain tract-based spatial statistics (TBSS) analysis.

• Supplementary file 2. Results of the mediation analyses across the three ROIs.

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
