## [Decision Letter]

Thank you for submitting your work entitled “Dissociable roles of the inferior longitudinal fasciculus and fornix in face and place perception” for peer review at *eLife*. Your submission has been favorably evaluated by Timothy Behrens (Senior Editor), a Reviewing Editor, and three reviewers.

The reviewers have discussed the reviews with one another and the Reviewing editor has drafted this decision to help you prepare a revised submission.

The following individuals responsible for the peer review of your submission have agreed to reveal their identity: Jody Culham (Reviewing Editor) and Charan Ranganath (peer reviewer).

The reviewing team agreed that the paper is “exciting”, “well written”, and “important and timely”.

However, there are a number of aspects that need to be addressed in a revision before the paper could be considered for publication in *eLife*. Though *eLife* usually provides consolidated reviews when revisions are required, the reviewers' comments were quite extensive and are appended below so authors have full access to all the points raised.

Essential revisions:

1) The authors should reconsider the treatment of the surprising correlation between hippocampal deactivation and both fornix FA and task accuracy. In their reviews, Reviewer #1 suggested more explanation, while Reviewer #2 suggested that the results should just be presented as “surprising” without too much handwaving. Upon post-review discussion amongst reviewers, the consensus was that the discussion should be retained but revised. Reviewer #3 suggested acknowledging the broader literature, noting that baselines are difficult to define for the hippocampus.

2) More discussion of the anatomical inputs to the hippocampus should be added (Reviewer #2, point 4).

3) A direct statistical test should be performed to test whether the brain-behavior correlations are significantly greater for the fine-grained perceptual tasks vs. the simple featural task (Reviewer #2, point 4).

4) Further justification is needed for the choice of regions based on probabilistic atlases rather than direct contrasts between faces and scenes (Reviewer #3, point 2). In addition, the authors should acknowledge that connectivity only correlates with the contrast of scenes vs. rest (not scenes vs. faces).

5) The reviewers wondered what happened in other areas including PPA (Reviewer #2, points 2 and 3) and other nodes of the scene network (RSC, TOS). Some suspected that these areas may have been analyzed but didn't show interesting patterns and thus weren't included. If so, a brief mention would be helpful along with an acknowledgment that the data for scenes and the fornix have less specificity than those for faces and the ILF. If not, the authors should consider including these analyses, since all three reviewers agreed in discussion that this analysis would be beneficial.

Reviewer #1:

In this study, the authors present evidence that various subregions of the MTL – the perirhinal cortex (PrC) and hippocampus (HC) – are selectively involved in complex face and scene perception, respectively, by virtue of their dissociable broader anatomical connectivity. Measures of white matter (WM) integrity of the inferior longitudinal fasciculus (ILF) and fornix were selectively correlated with performance on face and scene perception tasks, respectively. Further, BOLD response in the PrC (and FFA) and HC was selectively associated with face and scene perception, respectively. Finally, WM integrity of the ILF and fornix was associated with category-selective BOLD responses in PrC and HC, respectively, as well.

This is a very well written study, with sound methods and analyses that reports a strikingly consistent set of 3 double dissociations. While the total number of subjects is somewhat low for investigating brain-behaviour correlations (ranging from 24-29 across the various analyses), the pattern of results is very consistent across analyses of different measures and entirely consistent with the authors' hypotheses, lending credibility to the findings. Taken together, these results provide very compelling evidence that the PrC/ILF WM integrity and HC/fornix WM integrity play dissociable category-selective roles in face and scene perception, respectively. I have only one substantive concern:

The correlation between *deactivation* of the HC and both fornix FA and accuracy on the scene-oddity task being interpreted as evidence of HC involvement in complex scene perception is counterintuitive and entirely perplexing. The authors make the argument that a unique property of the HC, and the anterior portion of it in particular, is that increased neural activity causes a *decrease* in the BOLD signal, unlike neocortical regions and even other subregions of the MTL. The authors site a few select studies to support this possibility, including a number of studies that demonstrate reduced HC BOLD signal for various tasks relative to a resting-baseline. First, the latter studies (e.g., [108]; Wig et al., 2008) are probably not relevant here, in that the assumption in these cases is that ongoing cognitive processes during “resting-baseline” engage the HC to a greater extent than the constrained tasks of interest (i.e., the idea of HC involvement in the so-called “default mode”). Thus, in these cases, a decrease in BOLD activity presumably reflects a *relative* decrease in HC activity. Contrarily, the authors of the current study argue that scene perception drives HC activity, which results in a reduced BOLD response. To support this possibility, the authors cite a review by Ekstrom (2009), and several studies reviewed therein, that purportedly show decreased BOLD signal accompanying increased LFP and/or neuronal firing rates. However, such results are not robust or consistent, and are typically found under very specific and often highly atypical (e.g., epileptic activity) circumstances. Rather than concluding that the BOLD signal is negatively correlated with neural activity in the HC, as suggested by the authors, Ekstrom (2009) states that: “BOLD changes in the hippocampus correlated weakly or not at all with LFP power changes. We did not find a significant relationship between BOLD activity and neural firing rate […] our results suggest that the BOLD signal in the human hippocampal area has a more heterogeneous relationship with underlying neural activity than has been described previously in other brain regions,” (quoted from the abstract). Thus, the discussion of this work was somewhat misleading. Moreover, the suggestion that engagement of the HC results in reduced BOLD response is at odds with the vast corpus of cognitive neuroscience literature over the last two decades concerning the role of the HC in cognition. While there are far too many studies that report increased BOLD signal in the HC as a reflection of HC activity to list here, one such example is a previous study (75) with the same last author as the current paper, that appears to completely contradict this notion. In the [75] paper, increased BOLD response is reported as a measure of HC activation throughout (see Figures 6, 7, and 8) – indeed, the abstract of this paper states: “[fMRI] Functional region-of-interest analyses revealed that posterior HC and posterior parahippocampal gyrus showed greater activity [i.e., increased BOLD signal] during scene pattern learning, but not face and dot pattern learning, whereas PrC, anterior HC, and posterior fusiform gyrus were recruited [i.e., increased BOLD signal] during discrimination learning for faces, but not scenes and dot pattern learning. Critically, activity [i.e., increased BOLD signal] in posterior HC and PrC, but not the other functional region-of-interest analyses, was modulated by accuracy (correct > incorrect within a preferred category).”

How do the authors reconcile the current findings with these seemingly contradictory previous findings by (some of) the same authors?

Reviewer #2:

In this paper, Hodgetts and colleagues use diffusion tensor imaging (DTI) to characterize pathways involved in scene and face processing in young adults. The authors find that the structural integrity of the fornix is predictive of performance on the scene perceptual “oddity” judgments, and that the inferior longitudinal fasciculus is predictive of performance on face perception tasks. Using fMRI, the authors demonstrate that BOLD signal in hippocampus and perirhinal cortex is correlated with accuracy of scene and face oddity judgments. Finally, the authors find that the integrity of the fornix and ILF predicts BOLD responses in the hippocampus and perirhinal cortex, although the effects in the fornix are in the opposite direction to what would be expected.

The results from this study are exciting, in that they go beyond identifying simple dissociations between medial temporal lobe regions and instead relate them to networks that contribute to both memory and perception. Furthermore, it is striking that such large effects can be observed in healthy young individuals, potentially speaking to the utility of these behavioral and connectivity measures as early biomarkers for processes that can be affected in neurodegenerative diseases. I have some comments and concerns that are significant – particularly with respect to the results for the fornix – but I think that they can be addressed:

1) I was impressed with the statistical rigor and straightforward presentation of the data, with one significant exception – the treatment of the correlations between WM integrity and BOLD responses in the fornix. The authors initially present clear predictions about WM and BOLD, but then present a post-hoc argument that tries to make the case that we should actually expect fornix integrity to correlate with deactivations during scene oddity task performance, rather than positive activation. I am aware of the Ekstrom paper that the authors use to motivate their analysis, but if one takes his argument seriously, then one needs to discard or reinterpret any fMRI study that reports hippocampal activation that is positively related to perceptual or mnemonic processing (e.g., [65]). Using this argument really damages the authors' credibility, because the paper certainly does not take the Ekstrom argument into account when interpreting previous fMRI evidence for perceptual processing in the medial temporal lobes. I don't think that the authors need to make sense of this result, and that it would be far preferable for the authors to simply report the direction of the correlation and admit that it's surprising.

2) Related to point 1 above, it would be helpful for the authors to show voxel-wise and ROI analyses of fMRI data showing the contrast of face- vs. scene-oddity trials. This information will provide essential context for understanding the correlational data. It would be useful to know whether the parahippocampal place area (PPA) and hippocampus show increased activation during scene oddity trials.

3) Speaking of the PPA, it is surprising that the authors did not mention the PPA, which has been strongly implicated in scene processing and discrimination. In fact, the authors need to discuss why ILF integrity would be expected to correlate with face but not scene oddity. My understanding is that this white matter pathway would be expected to provide information to both the FFA and PPA.

4) A major strength of DTI is that it provides important information about anatomy, but the current manuscript does not provide an adequate treatment of the anatomy. For instance, the authors do not clearly describe the visual areas that provide input to the PRC via the ILF. Neuroanatomical context is especially important for motivating the authors’ predictions about the fornix, because it is not clear how the Saksida-Graham model would explain these results. The computational model describes a hierarchy of visual inputs that converge in the PRC and hippocampus, but I do not see how information about visual features would be conveyed via the fornix. The functions of diencephalic inputs are poorly understood, but from what I understand we would expect them to send information about head direction, not features of the visual scene. I think the paper needs to present a clear treatment of the anatomical input that would be conveyed to the hippocampus via the fornix and clarify how this input would be related to the kind of hierarchical visual processing that is proposed in their theoretical framework.

5) The study makes the case that the white matter measures are correlated with performance on the fine-grained perceptual tasks, but not on the simple featural tasks. However, the latter conclusion is simply a null effect, and it is not clear whether the correlations are significantly larger for face or scene oddity than for the size oddity task. These comparisons would bolster the author's conclusions. Alternatively, it would be useful to know if WM correlations with face and scene oddity hold up when you control for the influence of size oddity performance (i.e. this would show that white matter integrity is predictive of face/scene oddity over and its contribution to lower level visual discriminations).

Reviewer #3:

The current study was designed to test predictions derived from a representational account of MTL functioning. This model emphasizes differences at the level of stimulus content represented in perirhinal cortex (PrC) and the hippocampus (HC) that are critical for performance on tasks (mnemonic or perceptual) that require fine-grained discrimination. Specifically, PrC has been proposed to represent complex object information, including faces, whereas the HC purportedly plays a corresponding role for complex scene information. The prediction tested here is that PrC and the HC show distinct connectivity with other brain regions related to performance on perceptual discrimination for such stimuli. The authors examined the relationship between measures of WM microstructure of two fibre tracts, namely the inferior longitudinal fasciculus (ILF) and the fornix, task-related activity in the PrC and HC, and performance on an oddity task for faces and scenes. The overall pattern of results suggests links between activity in PrC, the ILF and performance on the face discrimination task, as well as links between the HC, fornix, and performance on the scene task.

The paper addresses important and timely questions about the functional roles of different MTL structures. The focus on connectivity in combination with focal task-related activity is to be commended given recent progress in understanding of category-specific responses in the ventral visual pathway based on similar types of analyses (e.g. [43]). Also, the matching of task requirements and performance across stimulus categories is a noticeable strength of the current investigation. However, there are a number of issues that relate to the selection of fibre tracts, regions of interest, and methods of analyses that make interpretation of the presented results somewhat difficult.

1) The authors motivate their focus on connectivity with reference to placing category-related functional dissociations between MTL regions into the broader context of distributed brain networks with similar content specificity. The focus on ILF as a marker of connectivity between PrC and the FFA is easily justified, given recent discussions as to the unique contributions of these regions to face processing. However, it's not immediately clear how a focus on the fornix can accomplish similar goals with respect to scene processing. Regions with scene specific responses have also been reported in posterior cortical structures, including the parahippocampal place area (PPA), retrosplenial cortex, and the transverse occipital sulcus (see Nasr et al., NeuroImage, 2013). I was surprised that these regions are not given any consideration in the current study. The authors note that the fornix connects the HC to diencephalic and other cortical structures, but what these structures are, and how they might relate to category-specific processing is not considered. Work by Aggleton and colleagues suggests that cortical targets of the fornix are found extensively in the frontal lobe; it is unlikely that such frontal regions would contribute to scene processing in a manner similar to the FFA's role in face processing. A more revealing analysis for this comparison might focus on connectivity between the HC and the PPA, retrosplenial cortex, and/or the transverse occipital sulcus in relation to performance on the scene discrimination task.

2) Although the authors cite a number of studies that have linked activity in PrC and the HC to perceptual discrimination of faces and scenes, respectively, they do not report corresponding results for their critical contrasts, without concomitant consideration of connectivity (see Figure 3) in the current study. I realize that demonstrating this link isn't the primary objective, but it seems important to first establish this pattern of results by contrasting F>S and S>F, irrespective of WM measures. On a related note, it is unclear why the authors used a probabilistic atlas, rather than an ROI based on this experimental task contrast, to define the FFA in the current study.

3) The analyses that combine consideration of focal task related activity for scene processing with structural connectivity only implicate the fornix when the contrast is between the scene-discrimination task and rest, rather than between the scene-discrimination task and the face processing task. This result seems to provide only limited support for a stimulus specific involvement, in particular given the emphasis the authors place on matching of task requirements and performance in such comparisons.

[Editors' note: further revisions were requested prior to acceptance, as described below.]

Thank you for resubmitting your work entitled “Dissociable roles of the inferior longitudinal fasciculus and fornix in face and place perception” for further consideration at *eLife*. Your submission has been evaluated by Timothy Behrens (Senior Editor) and Jody Culham (Reviewing Editor).

You've addressed almost all the points well. There's just one small revision the Reviewing Editor would like to request. Regarding effects in RSC, PPA, and TOS, the previous decision stated: “Some [reviewers] suspected that these areas may have been analyzed but didn't show interesting patterns and thus weren't included. If so, a brief mention would be helpful along with an acknowledgment that the data for scenes and the fornix have less specificity than those for faces and the ILF.” You've now done the analysis and found no significant effects but it doesn't seem to be mentioned in the manuscript. Because it was a common question from multiple reviewers and you did the work of the analysis, please add a brief mention it in the manuscript.

---

## [Author Response]

1) The authors should reconsider the treatment of the surprising correlation between hippocampal deactivation and both fornix FA and task accuracy. In their reviews, Reviewer #1 suggested more explanation, while Reviewer #2 suggested that the results should just be presented as “surprising” without too much handwaving. Upon post-review discussion amongst reviewers, the consensus was that the discussion of should be retained but revised. Reviewer #3 suggested acknowledging the broader literature, noting that baselines are difficult to define for the hippocampus.

To address these points, the relevant section in the manuscript has been rewritten as follows:

“The reported association between fornix microstructure (FA and to a lesser extent MD) and scene-related BOLD activity in HC was in the opposite direction to that observed between ILF microstructure and face selectivity in PrC/FFA, with fornix microstructure positively correlating with HC scene *deactivations*. […] One possibility, therefore, is that sparser blood supply in the HC (e.g., lower capillary density; [20]) leads to a decoupling between neuronal activity and BOLD, that is, where oxygen metabolism exceeds local blood flow.”

2) More discussion of the anatomical inputs to the hippocampus should be added (Reviewer #2, point 4).

In response to Reviewer 2, we have revised the manuscript as follows:

“[…] the reciprocal interplay between HC and surrounding neocortical and subcortical regions (3; 100) – that is afforded partly by fornical connections – appears critical for the formation of flexible spatial representations in the HC (i.e., those that maintain the coherent layout of a spatial environment across multiple viewpoints). […] Interestingly, this may also account for the moderate, though non-significant, association between size oddity and fornix microstructure (see Results).”

3) A direct statistical test should be performed to test whether the brain-behavior correlations are significantly greater for the fine-grained perceptual tasks vs. the simple featural task (Reviewer #2, point 4).

Direct statistical tests (Steiger z-tests) comparing the brain-behaviour correlations for (1) scene vs. size, and (2) face vs. size, were initially presented in [Supplementary-material SD4-data]. These are still reported and have now been moved into the main manuscript:

“While none of the microstructural measures obtained, in either pathway, were significantly associated with performance in the difficulty-matched size oddity condition, there were, as reported above, small-to-moderate one-tailed trends between fornix/ILF MD and size oddity (Figure 1—figure supplement 1). A Steiger Z-test comparing these coefficients revealed a significant difference between the face and size oddity correlation for ILF MD (z (26) = 2.05, p = 0.02). The difference between the size and scene oddity correlations for fornix MD did not differ significantly (z (26) = 0.94, p = 0.17).”

Based on the suggestion by Reviewer 2 (point 4), we conducted partial correlations to see whether the significant relationship between face/scene oddity and fornix/ILF MD remains when size oddity is controlled for – i.e. to show that white matter microstructure is predictive of face/scene oddity over and above its contribution to lower-level visual discriminations. When size oddity is controlled for, we still observe significant associations (one-tailed) between scene oddity and fornix MD (r = -0.38, p = 0.02, 95% CI [-0.61, -0.08]), and face oddity and ILF MD (r = -0.53, p = 0.00, 95% CI [-0.61, -0.08]). This additional analysis is now reported in the Results section.

4) Further justification is needed for the choice of regions based on probabilistic atlases rather than direct contrasts between faces and scenes (Reviewer #3, point 2). In addition, the authors should acknowledge that connectivity only correlates with the contrast of scenes vs. rest (not scenes vs. faces).

The reason for this is three-fold: firstly, probabilistically defined ROIs were used to ensure consistency in the analyses conducted between regions. Defining functional ROIs (while preferable in certain contexts) would have inevitably led to variability in the contrasts and thresholds used to identify clusters in individual subjects, particularly given variation in signal-to-noise across medial temporal lobe and temporo-occipital fusiform regions. This leads us to the second reason, namely that an anatomical ROI approach avoids the loss of individuals from our analysis (see also [65]; [10]; [75]), which would reduce the overall power of the experiment, and would also lead to potential bias in the analysis by only including those participants that, for example, show significant BOLD increases for faces versus scenes. Finally, this method was chosen to ensure non-circularity in our BOLD-behaviour and mediation analyses – i.e. we wanted to ensure that ROIs were independent of the BOLD signal change data that were analysed as part of the mediation analyses (Kriegeskorte et al., 2009).

The finding that fornix connectivity only correlates with scenes vs. rest (not scenes vs. faces) is now acknowledged in the Discussion section.

*5) The reviewers wondered what happened in other areas including PPA (Reviewer #2, points 2 and 3) and other nodes of the scene network (RSC, TOS). Some suspected that these areas may have been analyzed but didn't show interesting patterns and thus weren't included. If so, a brief mention would be helpful along with an acknowledgment that the data for scenes and the fornix have less specificity than those for faces and the ILF. If not, the authors should consider including these analyses. They are not strictly required as* eLife *avoids asking for extensive additional analyses unless they are necessary for the conclusions. However, since all three reviewers agreed in discussion that this analysis would be beneficial, the authors should consider including them.*

We did not look at this initially as our focus was on characterising the association between category-selective BOLD and microstructure in only those medial temporal ROIs that are directly connected to our tracts of interest. Based on the reviewers’ suggestion – that fornix microstructure may be associated with BOLD in other scene-selective cortical regions (RSC, PPA, and TOS) – we conducted an additional voxel-wise analysis within ROIs of the RSC, TOS and PPA. To be consistent with analyses presented in the manuscript, we used anatomically-defined, independent ROIs for the PPA (posterior parahippocampal gyrus from the Harvard-Oxford Cortical Atlas), RSC (Brodmann Area 29 dilated by a single voxel; see [19]) and TOS (a probabilistic mask from the ICBM Sulcal atlas; [68]). Using the same statistical thresholds presented in the paper, no significant clusters were found that showed a significant positive association between scene-selective BOLD (S > F, S > rest) and fornix microstructure (MD or FA) in any of the additional scene-selective ROIs.

[Editors' note: further revisions were requested prior to acceptance, as described below.]

You've addressed almost all the points well. There's just one small revision the Reviewing Editor would like to request. Regarding effects in RSC, PPA, and TOS, the previous decision stated: “Some [reviewers] suspected that these areas may have been analyzed but didn't show interesting patterns and thus weren't included. If so, a brief mention would be helpful along with an acknowledgment that the data for scenes and the fornix have less specificity than those for faces and the ILF.” You've now done the analysis and found no significant effects but it doesn't seem to be mentioned in the manuscript. Because it was a common question from multiple reviewers and you did the work of the analysis, please add a brief mention it in the manuscript.

A brief mention of this analysis has now been added to the Results and Methods sections of the manuscript. These results are described as follows:

“To test whether fornix microstructure is associated with scene-selective BOLD in other scene-selective cortical regions (34), we conducted an additional voxel-wise analysis within anatomically-defined, independent ROIs sampling the posterior parahippocampal gyrus (PHG), retrosplenial cortex (RSC) and transverse occipital sulcus (TOS; see Methods). No significant clusters were found that showed a significant positive or negative association between scene-selective BOLD (S > F, S > rest) and fornix microstructure (MD or FA) in any of the additional scene-selective ROIs.”